# Navigating the ERK1/2 MAPK Cascade

**DOI:** 10.3390/biom13101555

**Published:** 2023-10-20

**Authors:** Ana Martin-Vega, Melanie H. Cobb

**Affiliations:** 1Department of Pharmacology, UT Southwestern Medical Center, 6001 Forest Park Rd., Dallas, TX 75390, USA; ana.martinvega@utsouthwestern.edu; 2Simmons Comprehensive Cancer Center, UT Southwestern Medical Center, 6001 Forest Park Rd., Dallas, TX 75390, USA

**Keywords:** ERK1/2, MAPKs, cancer, scaffold, therapies, inhibitors

## Abstract

The RAS-ERK pathway is a fundamental signaling cascade crucial for many biological processes including proliferation, cell cycle control, growth, and survival; common across all cell types. Notably, ERK1/2 are implicated in specific processes in a context-dependent manner as in stem cells and pancreatic β-cells. Alterations in the different components of this cascade result in dysregulation of the effector kinases ERK1/2 which communicate with hundreds of substrates. Aberrant activation of the pathway contributes to a range of disorders, including cancer. This review provides an overview of the structure, activation, regulation, and mutational frequency of the different tiers of the cascade; with a particular focus on ERK1/2. We highlight the importance of scaffold proteins that contribute to kinase localization and coordinate interaction dynamics of the kinases with substrates, activators, and inhibitors. Additionally, we explore innovative therapeutic approaches emphasizing promising avenues in this field.

## 1. Introduction: MAPK Pathways

MAPK (Mitogen-Activated Protein Kinase) signaling pathways are among the best studied ubiquitous regulatory cascades that respond to a wide range of extracellular hormones, growth factors, neurotransmitters and other cellular cues to control rapid cellular responses, decision making and cell fate. MAPK cascades are comprised of three or more kinases acting in series, beginning with MAPK kinase kinases (MAP3Ks) that activate MAPK kinases (MAP2Ks or MAP/ERK kinases (MEKs)), that then with high selectivity activate MAPKs, the effector kinases of the cascades. Prior to its purification or recognition of its enzymatic activity, the MAPK ERK2 (extracellular signal-regulated kinase 2) had been observed as a 42 kDa tyrosine-phosphorylated protein in lysates from growth factor-stimulated and virally transformed cells [1]. Subsequently, what turned out to be the same protein was found as a growth factor-activated protein kinase [2] that phosphorylated microtubule-associated protein 2 (MAP-2 Protein Kinase) and could activate ribosomal protein S6 kinase [3]. Purification and molecular cloning identified two closely related but distinct enzymes ERK1 (p44, MAPK3) and ERK2 (MAPK1), often the more abundant [4,5,6]. Other MAPKs include Stress-Activated Protein Kinases (SAPKs)/Jun-N-terminal Kinases (JNKs), p38 family kinases and ERK5/Big MAPKs (BMK), in addition to atypical MAPK pathways ERK3/4, ERK7/8 and NLK (Nemo-like kinase) [7,8,9]. These MAPKs belong to the CMGC group of highly conserved serine–threonine kinases [10]. A hallmark of the typical MAPKs is their activation by dual phosphorylation on nearby tyrosine and threonine residues in their activation loop, in what is known as the TXY motif [11], where the amino acid X is distinct for subsets of MAPKs. In the case of ERK1/2 and ERK5, the motif is Thr-Glu-Tyr (TEY), JNKs contain Pro as the intermediate residue (TPY), and Gly for p38s (TGY) [9] (reviewed in more detail in [12]).

While all MAPK pathways handle an array of functions in cellular physiology and pathology, this review focuses on the ERK1/2 MAPK pathway. This pathway is crucial in maintaining normal functions in most if not all cells, and abnormalities in this pathway disturb many cellular processes and contribute to pathologies ranging from birth defects to cancer. As a consequence, significant efforts have been dedicated to understanding and manipulating this pathway. Here, we provide an overview of pathway components and functions and highlight the broad array of scaffolds as well as novel interventional strategies.

## 2. ERK Pathway Components

### 2.1. RAS

RAS (RAt Sarcoma) refers to small GTPase (G)-proteins encoded by three genes yielding proteins with molecular weights of ~21 kDa. RAS proteins contain a GTP-binding domain and a hypervariable region (HVR) which marks the following differences between them [13]: HRAS (Harvey sarcoma viral oncogene), NRAS (neuroblastoma oncogene) and KRAS (Kirsten sarcoma viral oncogene). KRAS has two alternatively spliced forms named KRAS4A and KRAS4B. To be active, RAS proteins need to be localized at the plasma membrane or endomembranes. Lipid post-translational modifications (PTMs), farnesylation and palmitoylation, are essential for RAS association on the membrane [14,15]. The membrane sub-domain and subcellular localization from which the RAS signal originates determines the downstream effectors that are activated and, consequently, the cellular outcome [16]. As GTPases, RAS proteins cycle from the GDP-loaded OFF configurations to GTP-loaded ON conformations assisted by RAS–guanine nucleotide exchange factors (RAS-GEFs) and return to OFF states facilitated by RAS–GTPase-activating proteins (RAS-GAPs) [17]. The RAS-GEFs SOS (Son of Sevenless)1 and SOS2 are activated by receptor tyrosine kinases coordinated by the adaptor protein Grb2 [18]. Once activated, RAS can interact with the RAS binding domain (RBD) in RAS effectors. In addition to RAF proteins, MAP3Ks in the ERK1/2 pathway, there are over fifty other RAS effectors including RALGDS, RALBP1, RIN1, TIAM1, PLCε, REPAC, RASSF1/5, and PI3K among others [19,20].

The four RAS proteins show around 85% of amino acid identity differing in the hypervariable region (HVR) [21]. In spite of their similarities, manipulation of their expression or activation results in different cellular outcomes [22,23]. KRAS is the only family member essential for development [24,25,26]. This feature depends on its locus expression pattern in development, as the protein product of HRAS knock-in in the KRAS locus gives rise to viable mice. However, a phenotype in adult life has been observed when KRAS is replaced by HRAS [27], although it disappears by eliminating the endogenous HRAS [28]. These findings indicate that different RAS isoforms are apparently equivalent in physiological conditions. Nevertheless, activating mutations in the different RAS proteins lead to different outcomes in a tissue-specific manner from senescence to tumorigenesis. Moreover, the same activating mutation in KRAS and HRAS generates different types of tumors in adult mice [28].

Mutations in RAS isoforms are responsible for nearly 30% of all human cancers. KRAS exhibits the highest mutation frequency in cancer, present in ~85% of cases. NRAS at 11%, while HRAS accounts for only ~3% of all RAS mutation-dependent cancers. The most frequently mutated codons are G12, G13 and Q61 regardless of isoform. Their frequency varies, with codons 12 and 13 more commonly mutated in KRAS, while 61 is more commonly mutated in NRAS and HRAS [29]. Only now beginning to be understood functionally, the mutational frequency of the isoforms differs across different types of cancer; in descending order of prevalence: pancreatic adenocarcinoma (>60% KRAS), colorectal adenocarcinoma (under 40% KRAS, ~5% NRAS), lung adenocarcinoma (~30% KRAS), and skin cutaneous melanoma (~28% NRAS), among the most common cancers bearing RAS mutations [30].

Despite the high amino acid identity between RAS isoforms, their nucleotide sequences are less so [31]. The Counter laboratory proposed codon bias as a mechanism responsible for the different expression and oncogenicity of KRAS and HRAS. Even though KRAS is the most commonly mutated isoform, it is the most poorly translated due to rare codons. This group suggested that the potent oncogenicity of HRAS lead to cell cycle arrest and senescence, while the different codon usage of KRAS overcomes this issue becoming the most frequently mutated RAS in cancer [31]. In fact, they later demonstrated that substitution of KRAS rare codons for common codons rendered tumors less aggressive [32]. Following studies corroborated that rare codons in KRAS affect transcription due to histone modification and transcriptional activation recruitment, mRNA translation, and even protein conformation, highlighting the importance of codon usage [33]. Furthermore, the correlation between tRNA expression, codon usage, and oncogenicity was further validated not only within RAS isoforms but also across other families of oncogenes [34].

As previously pointed out, the mutational pattern varies among the different types of tumors, with certain isoforms, hotspot codons, and the specific mutation, more frequent depending on the tissue [35]. A possible explanation for the mutation prevalence, as suggested by the Counter laboratory, is that each mutation generates a different level of activity and each tissue establishes its specific ‘sweet spot’ within which this activity drives tumorigenesis, while above or below that range of activity transformation is not possible [36]. Furthermore, RAS mutation tropism may be the result of mutational specificity and selection as observed with the carcinogen urethane. Urethane shows a mutation tropism towards Q61L/R mutation in KRAS in the lung. However, if KRAS rare codons are replaced by common codons, to compensate for the resulting stronger activity, urethane will induce mutations in G12 instead as these are biochemically less active [37,38].

### 2.2. RAF

RAF (Rapidly Accelerated Fibrosarcoma) is the first kinase in the pathway core, serving as the MAP3K downstream of RAS [39]. The regulatory paradigm has it that RAF is cytosolic in resting conditions and translocates to the plasma membrane to be activated by RAS upon growth factor stimulation [40,41,42] (reviewed in [43,44]). In mammals, the RAF family consists of ARAF (65 kDa), CRAF (65 kDa) (aka RAF1), and BRAF (84 kDa, larger due to additional BRAF-specific sequence at its N-terminus) [45], which are ~75% identical in shared segments. Also in the family are the pseudokinases KSR1 and KSR2 discussed in Section 4.2.

RAF consists of a RAS binding domain (RBD) and a cysteine-rich domain (CRD), both in the N-terminal regulatory region of the protein and the kinase domain in the C-terminus. RAF was formerly divided into three conserved regions named CR1, CR2, and CR3. CR1 contains the RBD and the CRD, CR2 is a serine/threonine-rich domain, and CR3 refers to the kinase domain (reviewed in [46]). CRD facilitates RAF localization at the plasma membrane [47,48] and together with RBD both interact with RAS [49,50]. Furthermore, the kinase domain is predicted to directly interact with the plasma membrane [51]. In quiescent cells, the N-terminal regulatory region interacts with the kinase domain as an autoinhibitory mechanism, preventing RAF activation. Besides the inhibitory intramolecular interactions, a 14-3-3 dimer binds to two phospho-sites, S365 and S729 in BRAF, stabilizing its inactive monomeric conformation. Active RAS recruits RAF to the plasma membrane opening the monomeric conformation [52]. The S365 (in BRAF; S259 in CRAF) inhibitory site is exposed and the ternary complex SHOC2-MRAS-PP1 dephosphorylates it, releasing the autoinhibitory conformation [53,54,55]. In the meantime, one of the 14-3-3 protomers remains bound to the pS729, while the other one is available to bind another RAF molecule stabilizing an active RAF dimer [42,56]. Other molecules such as KSR and MEK participate in RAF activation, adding to its complex nature [57,58]. The primary substrates of RAF are the MAP2Ks MEK1 and MEK2 [59].

Of the three isoforms, BRAF is the most frequently mutated in cancer. BRAF mutations are present in 7.7% across all cancer types [60]. Overall, BRAF mutation prevalence in human cancers is almost 100% for hairy cell leukemia, 50% for melanoma, 45% for papillary thyroid cancer, 10% for colon cancer, and 10% for non-small cell lung cancer cases [61]. The most frequent mutation in BRAF is V to E substitution at residue 600 (V600E), accounting for 90% of BRAF mutations [62]. Around 200 low-frequency cancer-associated mutations have been described in BRAF [63]. Most of them are believed to reverse the autoinhibitory conformation in the absence of upstream signaling [64]. BRAF mutations have been classified into three groups as follows: class I mutations are those that affect BRAF V600 position, most commonly V600E but also V600K/D/R/M which are able to transduce the signal downstream as monomers; class II mutations function as RAS-independent dimers with reported increased kinase activity but still weaker than BRAF V600 mutants; and class III mutants show impaired kinase activity and do not directly phosphorylate MEK but retain the ability to bind RAS and heterodimerize with CRAF [65,66]. BRAF fusions have also been found which lack the inhibitory N-terminal domain leading to constitutive BRAF dimerization and are considered in class II [67].

### 2.3. MEK

Downstream of RAF are the two ERK-specific MAP2Ks [68,69], MEKs 1 and 2 of 44 and 45 kDa, respectively [70]. The MEK kinase domain contains an ERK-selective binding site near the N-terminus and a nuclear export sequence (NES) that favors its cytosolic localization [71,72]. MEK1 is activated by phosphorylation of two activation loop serine residues (S218 and S222, S222 and S226 in MEK2) [73]. Apart from canonical activation by RAF, MEK can also be activated by other MAP3Ks, including MOS [74], MAP3K1/MEKK1 [75], MAP3K8 (aka Tpl2/Cot) [76], and MAST1 [77], linking activation to distinct upstream stimuli (Figure 1). In addition, PAK1 phosphorylates MEK1 on S298 to increase the interaction with ERK [78,79,80], and it was found later that it was also able to phosphorylate MEK1/2 in the two canonical serines of the activation loop by the canonical activation of serine [81].

MEK1 is retro-phosphorylated by ERK at T292 [82]. This phosphorylation event is specific to MEK1 and enhances the accessibility of phosphatases facilitating the dephosphorylation of S218 and S222 in the activation loop [80]. As MEK1 and MEK2 form heterodimers and this residue is exclusive to MEK1, in the absence of MEK1, MEK2 is not subjected to negative regulation, thus leading to ERK-sustained activation via MEK2 [83]. As in the case of ERK1/2, MEK1 and MEK2 seem to play non-redundant roles in development [84,85]; however, this may be due to a dosage effect [86]. Nevertheless, some studies emphasize specific functions at least in some settings [87,88,89]. MEK acts as a cytoplasmic anchor for ERK, binding to it in the cytoplasm and mediating its export from the nucleus via its XPO1/CRM1, initially demonstrated by its sensitivity to leptomycin B [90,91].

MEK mutations are rare, found in less than 1% of all human cancers, and not concentrated in a specific region or codon [63], in contrast to the upstream activators RAF and RAS. Mutations in MEK have been functionally classified into three groups. (1) RAF-independent mutations consist of deletions that result in autophosphorylation in the activation loop, resulting in a potent downstream activation. (2) RAF-regulated alterations lead to some basal activation which is potentiated in the presence of RAF. Some of these mutations have been found in resistant tumors. (3) RAF-dependent mutations display the lowest activity of all the mutants and depend on RAF to be active acting as signal amplifiers [92].

### 2.4. ERK

ERKs contain the usual two-domain protein kinase structure (reviewed in [93]). The N-terminal domain binds ATP and contains an essential ATP-binding lysine, often referred to as the catalytic lysine. Protein substrates and regulators bind across the C-terminal domain as discussed more fully below. ERKs also include a MAPK insert in the C-terminal domain and other regions that are important for MEK recognition [94,95] and an essential C-terminal segment that extends across both domains (Figure 2).

Activation of ERK by MEK results from the phosphorylation of tyrosine followed by threonine in the TEY sequence within the activation loop [11,96,97,98]. Following phosphorylation by MEK, this C-terminal segment aids in repositioning alpha helix C to achieve an activated ERK conformation [98]. The combined phosphorylation of both residues increases ERK activity 50,000-fold [99].

The similarities and overlapping substrate specificities of ERK1 and ERK2 have fueled the long-running debate over their functional specificity [100,101,102,103]. Genetic studies using knockout mouse models showed that the deletion of either ERK1 or ERK2 led to different phenotypic outcomes. Although loss of either ERK gene impairs neuronal development [104], ERK1−/− mice are viable and fertile [105], while whole body ERK2 knockout results in embryonic lethality regardless of ERK1 expression [106]. Other work has reported divergent roles in specific contexts [107,108,109,110,111,112]. Phenotypic differences have been attributed to the allele number and differential expression of each isoform. Ectopic overexpression of one isoform compensated for the lack of the other, supporting the conclusion of redundancy [100]. It has been suggested that variations in the expression of either isoform will be compensated for to maintain a similar amount of total ERK [103]. A detailed analysis of ERKs across many species demonstrated that two *ERK* genes first appeared in bony vertebrates by a gene duplication, although both genes are not always expressed in all tetrapod species [102]. Phylogenetic analysis indicated that *ERK1* is evolving faster than *ERK2*, probably related to the much smaller size of the *ERK1* gene and consistent with the complex regulation of *ERK2* expression suggested by work in *Drosophila* [113].

N-terminal alanine-rich (NTAR) sequences in ERK1/2 are now known to improve their translation. NTAR sequences ensure precise translation and prevent frameshifts by reducing the speed of elongating ribosomes. Human ERK1/2 possess multiple Ala codons immediately downstream of the AUG codon with an additional in-frame start codon also present. Notably, most of these Ala codons are rare codons that are repeated in tandem. Collectively, these features optimize accurate translation minimizing errors, which reflects the need for proper ERK protein synthesis within cells [114].

#### 2.4.1. ERK Substrate Recognition

ERKs can interact with hundreds of substrates in any location in the cell [119,120]. In addition to the many phosphorylation sites that have been identified by examining defined targets, many more sites in putative effectors have been predicted by in silico approaches with some validated in vivo [121,122]. ERK1/2 show a preference for sites in the consensus phosphorylation sequence, PX(S/T)P, with a proline in the P+1 residue, following the phosphorylation site, and are thus considered proline-directed kinases. Many substrates contain only a minimal (S/T)P sequence. Unlike many other kinases, in MAPKs the pocket for the P+1 residue in the substrate is shallow creating a preference for proline at the position following the serine/threonine phosphorylation site [98,123]. Glycine and rarely alanine can also be accommodated at the P+1 site.

The huge number of phospho-sites found in proteins [121] raises the question of how specificity is achieved. While scaffolds (see Section 4.2) can form three-way complexes with enzymes and substrates, docking sites are important MAPK recognition mechanisms [124,125]. The best characterized docking sites identified on ERK are the common docking (CD) or D-recruitment site (DRS) and the F-recruitment site (FRS) or docking site for ERK FXF (DEF) (see Figure 2) [126]. The CD site, accessible in both inactive and active ERK configurations, binds a basic/hydrophobic motif, typically K/RX_2-4_LXL, referred to as a D motif, kinase interaction motif (KIM), or more generally as a short linear motif (SLiM). In addition to many substrates, MEK1/2 also bind ERK1/2 through a CD–D motif interaction, as does the MAP kinase phosphatase DUSP6. The FRS site binds proteins containing FXF motifs, including nuclear pore proteins [127], and is more accessible after ERK activation [128,129]. Some ERK substrates, such as Elk1, contain both interacting domains and some appear to lack either motif [130,131,132]. When a protein possesses multiple (S/T)P residues, docking motifs may play a crucial role in guiding the phosphorylation towards a particular residue [133]. Additional strategies are being developed to identify interaction mechanisms, including a recent study expanding our understanding of CD–D motif interactions [134]. Moreover, targeting the DRS to disrupt protein–protein interaction with ERK interactors has already been assessed [135,136].

#### 2.4.2. ERK Localization

ERK1/2 are found in cytoplasmic-, nuclear-, and membrane-associated compartments, and their subcellular distribution defines access to substrates. Localization is dynamic and is influenced by the duration and nature of activating stimulation. Two ERK monomers can associate to form a free dimer in the cytoplasm to activate cytoplasmic substrates. ERK dimers can be assembled on scaffold proteins in specific subcellular localizations where the ERK monomer not associated with the scaffold will phosphorylate specific substrates [137]. Binding the FG repeat regions of nuclear pore proteins, nucleoporins (Nup) Nup214, Nup153, TPR, and others, allows passive entry of ERK into the nucleus [138,139,140]. A study by the Nishida lab showed numerous interactions of ERK2 with nuclear pore proteins with regulatory impact [141]. Nup-mediated facilitated diffusion allows rapid exchange of free ERK2 between the cytoplasmic and nuclear pools and reveals the great regulatory importance of binding partners with restricted localizations that can influence where ERK2 acts.

While active nuclear import mechanisms have been identified, ERK scaffolding, to retain the kinases in the relevant compartment, and ERK nuclear export through binding to exported partners, appear to dominate processes determining ERK location. As noted earlier, binding to MEK promotes ERK nuclear export and favors cytoplasmic retention. Dual specificity phosphatases also influence ERK location by related mechanisms. The phosphatase DUSP6 is primarily cytoplasmic and binds ERK through a D motif–CD domain interaction; DUSP6 also possesses an NES, promoting ERK nuclear export. In the nucleus, ERK associates with many transcription factors and is also retained by de novo-synthesized nuclear anchors, such as another phosphatase, DUSP5 [142] and Vanishin [143]. Chromatin association of ERK1/2 is widespread as deduced from chromatin immunoprecipitation [144,145] (reviewed in [146]). Cytoplasmic anchors include microtubules which may bind as much as one-third to half of ERK2 [147]. PEA-15 can also anchor ERK in the cytoplasm [148] and prevents ERK2 phosphorylation by MEK1. PEA-15 directly binds ERK2 through the MAPK insert, which is also required for a productive interaction with MEK [149]. Another cytoplasmic anchor is Sef1, which interferes with ERK nuclear translocation without affecting its activation state or cytoplasmic functions. Sef1 binds to MEK1 and prevents MEK–ERK complex dissociation [150].

#### 2.4.3. ERK in Cancer

In contrast to upstream cascade components, ERK1/2 are rarely mutated and oncogenic activity has not been unequivocally demonstrated. ERK mutations recognized as of 2020 have been thoroughly reviewed by Smorodinsky-Atias (2020) [151]. Most of these mutations have been generated in the laboratory based on ERK orthologs, protein structure and two ancestors as follows: one common to ERK1/2/5 with autophosphorylation capabilities and another common to ERK1 and ERK2 that lacks the autophosphorylation ability. A few of them rendered some activation but usually minimal compared to that achieved by MEK [152]. Yet, many mutations have been found in cancer patients. The Johanessen group created a library with almost 7000 missense ERK2 mutations and identified some of them as gain- or loss-of-function in tumor samples. The ERK2 missense mutants were also assessed for resistance to different pathway therapeutics [153].

An ERK mutation with potential oncogenic activities is an intrinsically active mutant ERK1 with the mutation R84S, initially found in a screen for ERK MEK-independent mutants in an ERK yeast ortholog [154]. This mutant was able to transform NIH3T3 cells [155] and was activated in developmental assays in a *Drosophila* cancer model, especially combined with the sevenmaker mutation [156]. However, the equivalent mutation in ERK2 did not replicate the transformation outcomes. As of September 2023, it has not been reported in human cancer samples (COSMIC database [157]). The only mutation found in that residue in patients is R84H, which showed resistance to ERK inhibitors [158].

According to the COSMIC database, as of September 2023, ERK1 accounts for 0.5% of all the cancer samples analyzed. These mutations can be found throughout the protein with no hotspots identified, most of them were found only in one single sample with a maximum of four cases [157]. Mutations in ERK2 are slightly more frequent, representing 1.2% of the tested samples. Alterations also spread across the whole protein; however, there is a hotspot in residues 321 and 322 within the common docking domain. In fact, the most common mutation is E322K, which occurs in only 0.1% of all the assessed samples but accounts for 7% of all ERK2 mutations found in cancer samples. Although the majority of the mutations in ERK1 and ERK2 do not follow a tissue distribution pattern, mutations in the CD site of ERK2 are found mainly in squamous cell carcinoma, most frequently in cervix, oral-esophagus, head and neck, lung and oral tissues [159]. E322K was originally identified in oral squamous cell carcinoma [160]. Interestingly, the other most frequent mutation in this hotspot, D321N [161], and E322K are both considered gain-of-function due to decreased interaction with phosphatases [162,163] and are MEK activation-dependent [164]; however, they display functional differences [163,164]. The equivalent mutations in ERK1 do not produce gain-of-function mutants [158].

## 3. ERK Functions and Context Specificity

ERK1/2 are critical for integrating extracellular ligand binding cues with intracellular conditions to elicit coordinated cellular responses such as exit from a stem cell state, proliferation, terminal differentiation, and tissue-specific differentiated functions [165,166]. As noted above, ERK2 is essential early in embryonic development, and variation in a single allele is associated with developmental and cognitive impairment [104,167,168,169]. ERK2 can phosphorylate hundreds of downstream targets, including more than half a dozen protein kinase substrates which expand its regulatory reach in many subcellular locations [122,141,170,171,172,173]. The ability to communicate directly and indirectly with such a breadth and distribution of downstream proteins results in both the intrinsic potential of ERK2 to carry out ligand-specific directives and also the malleability of ERK2-elicited responses to context, tightly linked to the cellular proteome expressed at any specific time [174,175,176,177,178]. This plasticity also makes cells vulnerable to diseases arising from disturbances in ERK2 regulation. The activities of ERK1/2 must be adjusted up and down during the cell cycle, as development progresses, and in cells of the adult animal. This fluctuation in ERK1/2 activity is required to maintain or exit from a stem cell state, to facilitate terminal differentiation, and to maintain cell identity. ERK1/2 activities are necessary participants that mediate acute endocrine, paracrine, and autocrine stimulation, and adjust outputs due to changes in nutrient availability and other cellular stresses to support the context-specified cellular responses. Given the broad variety of ERK substrates and their intricate interactions, ERK orchestrates many cellular functions in physiological conditions and diseases. While ERK functions have been reviewed extensively elsewhere [165], a concise overview of some aspects is provided here.

### 3.1. Cell Cycle, Growth, Migration, and Survival

ERK regulates G1 to S phase transition, and its depletion results in cell cycle arrest [179] (reviewed in [180]). Cell growth is essentially biomass accumulation due to anabolic growth pathways [181]. ERK participates in the regulation of several processes implicated in growth for the preparation of cell division (reviewed in [182]). For instance, ERK stimulates the de novo synthesis of pyrimidine nucleotides to generate DNA and RNA. ERK phosphorylates the CAD protein (carbamoyl-phosphate synthetase 2, aspartate transcarbamylase, and dihydroorotase), increasing pyrimidine biosynthesis and antagonizing the negative effect of PKA CAD phosphorylation [183]. Ribosomes are necessary to handle new protein synthesis. ERK stimulates ribosome biogenesis by phosphorylation of RNA polymerase I [184] and III [185]. Finally, ERK is involved in protein translation by regulation of mammalian target of rapamycin complex 1 (mTORC1), eukaryotic translation initiation factor 4E (eIF4E) through MNK1/2, and cytoplasmic polyadenylation element binding protein 1 (CPEB1) [186] (reviewed in [187]). ERK increases mRNA polyadenylation and translation [188].

ERK regulates many steps of cell migration including membrane protrusion, cell matrix adhesion and cell contraction. ERK signaling promotes epithelial to mesenchymal transition (EMT) [189] by regulating pro-motile and pro-invasion gene expression, such as TWIST1 [190], SNAI2/Slug, and ZEB1/2 [191], and through DOCK10-dependent Rac1/FoxO1 activation [192].

ERK has both pro-apoptotic and anti-apoptotic functions. In response to specific stimuli such as stress or signaling from the death receptors, tumor necrosis factor (TNF), Fas, and TNF-related apoptosis-inducing ligand (TRAIL) receptors, ERK protects against apoptosis [193] by phosphorylating BCL-2 (B cell lymphoma 2) family pro-survival proteins and promoting degradation of pro-apoptotic proteins such as BAD [194], BIM [195,196] and caspase 9 [197]. ERK-mediated phosphorylation of cAMP response element-binding protein (CREB) leads to the transcription of pro-survival genes, even though sustained ERK activation negatively regulates CREB [198]. ERK may have dual pro- and anti-apoptotic functions, as in the case of thymocytes, depending on the activating stimulus [199] (reviewed in [200,201,202]).

### 3.2. Context Dependence of ERK Functions in Pancreatic Beta Cells

In pancreatic beta cells, circulating glucose regulates insulin secretion and production. ERK1/2 are essential in nutrient sensing, and their activities rise and fall as a function of glucose concentration over the physiologic range, in parallel with insulin secretion [203]. Glucose metabolism triggers calcium influx and release from intracellular stores to activate ERK1/2. Calcium influx also activates the calcium-dependent phosphatase calcineurin, which is required for maximal ERK1/2 activation by glucose [204]. Calcineurin is required for activation of ERK1/2 by other stimuli that induce insulin secretion as well, including depolarization and glucagon-like peptide 1, but not by phorbol ester or insulin. BRAF is also a calcineurin substrate; calcineurin dephosphorylates threonine 401 on BRAF, which is a site of negative feedback phosphorylation by ERK1/2 [205]. The major calcineurin-dependent event in glucose sensing by ERK1/2 is the activation of BRAF in beta cells [206].

The composition and localization of substrates molds cell characteristics and defines the potential regulatory consequences of ERK activation [207,208,209]. Changing the concentration of ERK substrates in pancreatic beta cells clearly demonstrates how ERK substrate composition influences differentiated properties of cells. In a matter of hours, ERK activity switches from inducing insulin gene transcription, by phosphorylating the glucose-sensitive transcription factors NEUROD1, PDX1, and MAFA, to inhibiting insulin gene transcription through the hyperglycemic induction of the substrate C/EBP beta, a transcription factor that inhibits insulin gene transcription [174,203,210,211]. Changes in substrate concentrations are induced by metabolic signals, not by ERKs themselves. Nevertheless, ERK actions on substrates induced by hyperglycemia strongly promote beta cell dysfunction, demonstrating how ERK may enhance or interfere with normal physiological functions of differentiated cells. Similar cell context-specific properties allow ERK to interconvert from its common role as a cancer-promoting pathway to a tumor suppressor in small cell lung cancer (SCLC) [212,213,214,215].

### 3.3. ERK in Stemness

The requirement for ERK in stem cells depends on the pluripotency state of the cells. There are two states of pluripotency as follows: naïve, that resembles an early-stage embryo (pre-implantation) and primed stem cells, that evoke a later stage (post-implantation) [216]. Naïve pluripotent cells require LIF/STAT3 (leukemia inhibitory factor/signal transducer and activator of transcription 3) signaling, whereas primed pluripotency depends on FGF (fibroblast growth factor)/ERK and activin signaling [217]. ERK activation is dispensable for naïve but essential for primed pluripotency. In the case of mouse embryonic stem cells, the activation of the LIF/STAT3 pathway [218,219] and BMP (bone morphogenetic protein) triggering Id (inhibition of differentiation) genes are enough to maintain pluripotency. To avoid differentiation, they are cultured in the presence of LIF [218] and BMP4 [220]. In this setting, ERK antagonizes STAT3 signaling promoting differentiation [221], but BMP4 suppresses ERK signaling to sustain self-renewal through Smad activation and DUSP9 upregulation [222,223].

Meanwhile, GSK3 (glycogen synthase kinase-3) inhibits Wnt/β-catenin signaling, which is essential for maintaining self-renewal and inhibiting differentiation in mESCs [224,225]. mESCs require MEK and GSK3 inhibitors in the culture medium for stability; this combination is known as 2i-culture [226,227]. ERK activation is crucial in mESCs to end self-renewal and initiate differentiation [228,229]. Nevertheless, attempts to completely abolish ERK expression by genetic means in mESCs have resulted in cell death, which indicates that ERK activity is still required for survival in these cells although at a low level [230].

Stem cells from mice (mESCs) and human (hESCs) do not behave identically. mESCs are derived from pre-implantation embryos [231,232], while hESCs are acquired from post-implantation embryos. Thus, mESCs display naïve pluripotency [233,234,235], whereas hESCs belong to a primed pluripotent state [236]. Accordingly, they display striking differences regarding culture conditions, activated signaling pathways and gene expression profiles.

While ERK signaling destabilizes pluripotency in mESCs, it has the opposite effect in the human counterparts that require the activation of the FGF/ERK pathway as well as the presence of activin/transforming growth factor β-1 (TGFβ1) to preserve self-renewal and pluripotency [236,237,238]. Indeed, mESCs and hESCs have an inverse response to FGF/ERK signaling. In human cells, high levels of exogenous FGF-2 are enough to maintain pluripotency [239,240,241], and ERK2 and ELK1 cooperate to regulate stemness repressing differentiation genes by binding their promoters [242]. In contrast to mice, FGF inhibition leads to differentiation [240,243]. In summary, ERK has context-specific actions in stem cells at different states of pluripotency and from different species as a powerful example of the complexity of its actions.

### 3.4. ERK as an Allosteric Regulator

ERK possesses multiple functions independent of its catalytic activity. In addition to the many activity-dependent actions on transcription factors (e.g., Elk1, NeuroD1) noted above, ERK also influences gene expression transcription factors, co-activators, and co-repressors through direct interactions. For instance, ERK acts as a transcriptional repressor for interferon gamma (IFNγ)-induced genes as both ERK2 and C/EBP-β compete to bind GATE element, and only when C/EBP-β is bound are the downstream genes expressed [244]. ERK directly assembles on a promoter element of the IL4 gene, aiding in recruitment of transcription factors that initiate IL4 gene transcription. Association with the promoter and IL4 transcription induction is necessary for human Th2-cell differentiation [245]. ERK2 directly binds the MYC promoter and recruits CDK9 which binds and activates RNA polymerase II, resulting in ERK-induced MYC expression independently of ERK activity [246]. Another non-catalytic ERK function as a transcriptional activator occurs by disrupting the interaction of Rb with Lamin A. Activated ERK translocates into the nucleus and displaces Rb from Lamin A. Thus, Rb is phosphorylated by other kinases in the nucleoplasm which disturbs Rb inhibitory binding to EF2 in the DNA, inducing transcription and cell cycle progression [247].

ERK allosterically regulates certain other proteins, notably DUSP6, independent of its kinase activity. A D motif in the non-catalytic amino-terminal domain of DUSP6 binds the ERK CD site. This interaction stabilizes the active conformation of DUSP6 by inducing the closure of the general acid loop in the phosphatase [248]. Thus, ERK provokes a conformational change in the DUSP6 catalytic domain increasing its phosphatase activity to negatively regulate ERK pathway activation [249]. In a similar manner, ERK is believed to activate Topoisomerase IIα independently of its kinase activity, yet in this case it needs to be phosphorylated [250]. Recent studies indicate that ERK phosphorylates Topoisomerase IIα, and both inhibition of ERK2 kinase activity and knock down alters Topoisomerase IIα functions [251]. When phosphorylated, ERK2 allosterically activates poly (ADP-ribose) polymerase 1 (PARP1) through mechanisms that are independent of its catalytic activity. PARP1 activation is crucial in the cellular response to DNA damage and DNA repair mechanisms [252]. In these cases, the active conformation of ERK may be required even though activity is apparently not [253]. Non-catalytic regions of ERK are also being exploited as therapeutic strategies (see inhibitor section below).

## 4. ERK Regulation

### 4.1. ERK Negative Feedback Loops

Among its extensive substrates, ERK can bind and phosphorylate upstream members of the cascade to exert a positive or negative impact on the downstream signal. Negative feedback loops downregulate the signaling maintaining pathway balance. In addition to upstream components, ERK can phosphorylate scaffolds and adaptor proteins to rapidly suppress pathway activity, referred to as direct regulation. ERK can also phosphorylate transcription factors responsible for the expression of negative regulators, giving rise to a delayed but long-term response known as indirect regulation (reviewed in [254]).

ERK can phosphorylate multiple sites on BRAF and CRAF, some preventing their heterodimerization and others the interaction with RAS and localization at the plasma membrane [255,256] (Figure 3). In the case of BRAF, ERK also phosphorylates several residues with a negative outcome [205]. The conserved motif SPKTP, not present in the other RAF isoforms, is found in the C-terminus of BRAF, and it is also phosphorylated by ERK and implicated in the negative regulation of the pathway [257]. Three other residues in BRAF, T401, S750 and T753, are phosphorylated by ERK and prevent heterodimerization with CRAF [205,258].

ERK also can negatively regulate the scaffold protein KSR1, while ERK phosphorylates T260, T274, S320 and S443 in response to growth factor stimulation or RAS activation [259,260,261]. These events are proposed to contribute to BRAF–KSR1 dissociation and relocation of KSR1 from the plasma membrane to the cytoplasm [262].

Another ERK-regulated negative feedback loop targets the guanine nucleotide exchange factor SOS. SOS is phosphorylated by ERK at residues S1132, S1167, S1178 and S1193 to decrease binding affinity with Grb2 and, consequently, block SOS translocation to the plasma membrane [263,264]. RSK2 activated by ERK is responsible for SOS phosphorylation at S1134 and S1161 [265,266,267]. These phosphorylations generate a binding site for 14-3-3 proteins sequestering SOS and prevent its participation in RAS activation [268,269].

Indirect regulation by ERK leads to the de novo synthesis of negative regulators by transcriptional upregulation of dual-specificity phosphatases (DUSPs) DUSP1, DUSP2, DUSP5, DUSP6 and DUSP9 [270]. In addition, ERK phosphorylates DUSP1 to stabilize the protein and increase its half-life [271,272]. ERK increases the expression of sprouty (SPRY) genes [273]. Sprouty1 and sprouty2 proteins act upstream of RAS by binding the adaptor protein Grb2 and impeding its interaction with SOS [274]. SPRY4 inhibits the pathway by binding CRAF [275].

### 4.2. ERK Scaffolds

Many actions of the MAPK cascade are targeted to subsets of substrates through association with scaffolding proteins. Nevertheless, the concentrations and affinities of pathway components, which vary widely among cell types [276], are also important for pathway outputs. Early efforts to quantify relative amounts of these signaling molecules in HeLa and Cos-7 cells suggested that MEK is the most abundant protein of the pathway, followed by ERK, then RAS, and RAF is the least abundant in these cell lines [277]. More recent studies have suggested increases in the molecules moving down the cascade with ERK2 as the enzyme of highest abundance [278]. Numerous studies have arrived at affinities for pairs of components (Table 1). Of course, K_d_ values determined from in vitro experiments may not reflect the cellular value as a result of differences in concentrations of scaffolds and interacting proteins in cells. Affinity predictions under physiological conditions are approximations at best [279]. Table 1 summarizes data from multiple in vitro studies.

Dozens of protein scaffolds have been linked to a function in the RAS–ERK pathway. These include adaptor or accessory proteins which are sometimes classified as scaffolds. Sticking to the benchmark that a scaffold binds simultaneously to at least two members of the same signaling cascade forming a functional, stable complex [292,293], in this review the discussion is limited to three types of the nearly 20 proteins that physically associate with at least two members of the cascade. Table 2 contains the current list.

The first scaffold characterized in a MAPK cascade was Ste5, identified in the budding yeast Saccharomyces cerevisiae. Ste5 binds Fus3 (a MAPK), Ste7 (a MAP2K) and Ste11 (a MAP3K) forming a multiprotein complex at the tips of mating projections following stimulation with mating pheromones [294,295]. Before the discovery of Ste5, it was generally assumed that kinases did not form stable complexes with substrates. Interestingly, no mammalian Ste5 homologue has been identified suggesting context specificity [296,297]. In contrast, scaffold proteins such as IQGAP [298] are evolutionary conserved from yeast to humans which suggests an essential cellular function.

Why are scaffolds important? Scaffolds are the least understood components of MAPK pathways, but they connect more directly to functional specificity than the individual cascade kinases themselves. For example, IQGAP1 [299], paxillin [300] and GIT1 [301] ERK scaffolds are responsible for localization of active ERK in focal adhesions and actin filaments. Here, ERK phosphorylates several substrates involved in actin polymerization, myosin activation and Rac/Cdc42 inactivation, among others (reviewed in [302]).

They incorporate diverse regulatory properties, as detailed below, in addition to serving as cascade assembly sites [303]. Scaffolds share a number of essential features ensuring enzymes like ERK1/2, which can phosphorylate hundreds of proteins, do so at the right place and time. Scaffold proteins optimize signaling by clustering enzymes and substrates together, increasing their effective concentrations. Scaffolds also position proteins in advantageous orientations, often to facilitate phospho-transfer reactions [304,305]. Other proteins or small molecules may allosterically regulate the pathway through a specific scaffold to increase or inhibit the cascade output [57,306]. Scaffolds can also sequester and protect the components of the cascade not only from dephosphorylation from phosphatases [293,305,307,308] but also from unwanted crosstalk, for example, by segregating a kinase involved in two cascades to the proximity of the components of one of them [309,310]. Obviously, this is not the case when two pathways, that cross-activate one another on purpose, share the same scaffold. In addition, scaffolds control the ERK signal in space; by localizing ERK activity in a specific subcellular localization [311,312], scaffolds dictate ERK substrate specificity [313]. Scaffolds can also work as dimerization platforms to promote the formation of ERK dimers. This function may also participate in substrate selectivity, as one ERK monomer binds to the scaffold and the other to a specific cytoplasmic substrate [314].

Also central to their function, scaffolds must be at an optimal concentration to produce maximum signal output. With too few scaffold proteins, cascade assembly may fall below the optimum, while too many scaffold proteins will separate the kinase cascade components, yielding unproductive complexes; in the latter case, more is less. This phenomenon has been named “combinatorial inhibition” [315] and the “prozone effect” [305,307,316,317]. Ideally, each scaffold should maintain the aforementioned optimum level that results in maximum signal efficiency. Nevertheless, to balance the necessary pathway outputs in a given cell context, these optimum levels may vary based on other scaffold concentrations that are competing for the same pool of kinases, as well as the upstream activating signal. For example, overexpression of a cytoplasmic scaffold enhances ERK activity in the cytoplasm and, as a consequence, downregulates nuclear ERK signaling, consistent with competition for the same pool of kinases [150,318,319,320]. Because of the dependence on scaffolds, regulating scaffold concentrations might be an effective method to control MAPK output, titrating activity down without fully suppressing it, perhaps with output selectivity. With a more precise understanding of scaffold function, such a strategy might offer an alternative to bypass resistance mechanisms. Modeling approaches have sought to fill this gap in our knowledge of scaffold behavior but have not yet developed models that have been validated experimentally [321].

Variations in the expression of a single scaffold have altered total ERK activity to a greater extent than just interfering with a single scaffold [322,323,324,325,326,327]. Furthermore, scaffolds can associate with each other to form macro-complexes. Interactions have been described between MP1 and MORG1 [328]; IQGAP1 and MP1 [329]; paxillin and GAB1 [330]; IQGAP1 and β-arrestin-2 [331]; and IQGAP1 and KSR1 [332]. It has been proposed that scaffolds coordinate and cross-talk [137]. This premise was substantiated a few years later by the functional interaction between KSR1 and IQGAP1. ERK bound to a KSR1 mutant unable to bind MEK can be transactivated from MEK bound to IQGAP1, and this cross-activation mechanism has been named trans-phosphorylation. This mechanism takes place in specific cellular settings, and it depends on the concentration of the scaffolds and proper stoichiometry among them and the kinases. Additionally, the downregulation of one scaffold can lead to the upregulation of another scaffold with compensatory abilities [332].

#### 4.2.1. KSR1/2

In 1995, KSR1 was simultaneously discovered in *Drosophila virilis*/*Drosophila melanogaster* [333] and *Caenorhabditis elegans* [334,335] by a genetic screen for RAS downstream effectors. The phenotypes of the KSR1 loss-of-function mutants revealed inhibition of RAS signals, and for this reason the protein was named Kinase Suppressor of RAS (KSR) [333,334]. KSR was recognized as a mammalian functional equivalent of the yeast MAPK scaffold protein Ste5, even though they do not share sequence homology.

The KSR family is composed of five members, the pseudokinases KSR1 [333] and KSR2 [336] and their relatives, the three RAF protein kinases (Figure 4). Despite close sequence similarity between KSR1 and KSR2, these two proteins have distinct functions which are due in part to differences in expression. While KSR1 is widely expressed, KSR2 has a restricted tissue expression pattern, generally with neuroendocrine enrichment. Regarding function, both are scaffolds of the RAS–ERK pathway, but the knockout phenotypes are markedly different. Mice lacking KSR1 show a mild phenotype with disorganized hair follicles but no developmental issues, while RAS-directed oncogenesis is reduced [323]. On the other hand, KSR2-deficient mice are less fertile and develop metabolic disorders such as obesity, insulin resistance and type 2 diabetes [337,338]. Structurally, KSR2 is larger than KSR1 [336], incorporating a distinct motif that binds the AMP-activated protein kinase, AMPK. Binding AMPK likely confers, at least in part, the unique association with metabolic disorders [337].

Why are KSR1/2 considered pseudokinases? Although mammalian KSR1/2 were initially classified as pseudokinases because they contain arginine in place of the lysine essential for catalytic activity, this has been a point of contention [348,349,350,351,352,353]. Some studies have reported residual kinase activity [306,351,354,355] not detected by others [342,356,357]. Indirect evidence suggesting catalytic function comes from the findings that ectopic expression of KSR1 or KSR2 is sufficient to induce proliferation in a RAS-independent manner, attributed to heterodimerization with RAF. Still, the exact mechanism by which KSR overexpression leads to ERK activation has not been defined [358]. Some work has reported that ATP binding is essential for KSR1 activation, although how KSR1 binds ATP was not addressed [306,358]. Adding to the puzzle, unlike mammalian KSRs, *Drosophila* and *C. elegans* KSRs possesses the same catalytic sequence as conventional protein kinases (VAVK) [344], raising the question of why arginine replaced lysine in the mammalian proteins. On the other hand, mutation of the ATP-binding lysine in *Drosophila* and *C. elegans* KSR does not compromise the function of either; in these organisms, mutant KSR still supports RAF activation and increases ERK signaling [356,357], consistent with a non-catalytic function.

KSR1 is usually cytoplasmic and thought to be constitutively bound to MEK [345,346,347,359]. Upon stimulation and RAS activation, KSR1 rapidly translocates to the plasma membrane where it interacts with a RAF isoform, especially with BRAF [58]. At the plasma membrane, KSR exhibits selectivity towards defined microdomains, responding preferentially to RAS signals from lipid rafts [339]. Interaction with the E3 ligase IMP (Impedes Mitogenic signal Propagation) promotes the recruitment of KSR to Triton-resistant structures that sequester KSR1 and block ERK activation [360,361]. RAS activation leads to IMP proteasomal degradation, facilitating KSR-mediated ERK activation [362]. In this respect, it has been shown that KSR selectively couples RAS signals from lipid rafts to the activation of cPLA2 by ERK [16]. The ERK-mediated KSR1 phosphorylation sites (Section 4.1) are proposed to contribute to BRAF–KSR1 dissociation and relocation of KSR1 from the plasma membrane to the cytoplasm [363]. Phosphorylation of KSR1 on S297 and S392 by the kinase C-TAK1 creates 14-3-3 binding sites; these sites must be dephosphorylated, most likely by PP2A, to localize KSR1 to lipid rafts [259,364,365]. KSR1 phosphorylation may be associated with its nucleo-cytoplasmic shuttling [366]. Fewer sites have been reported in KSR2 and none are well characterized, but some may be controlled by calcineurin dephosphorylation [255].

Homo- and heterodimerization of KSR and RAF proteins underscores the remarkable complexity of ERK regulation at the MAP3K level, some of which was originally uncovered through studies of RAF inhibitors. A depth of structural data has revealed multiple dimeric structures of these molecules [57,306]. KSR and RAF can form side-to-side heterodimers believed to trigger RAF activation through a nearly identical dimer interface that is conserved across all family members. In the same way, KSR1 can also homodimerize through its C-terminus, forming side-to-side dimers as well [57,367]. Apart from KSR–RAF interaction through the kinase domains, selective heterodimerization of RAF with KSR1 occurs through direct contacts between the N-terminal regulatory regions of each protein, including the KSR coiled-coil–sterile alpha motif (CCSAM) [340]. Interestingly, MEK binding to the kinase domain of KSR1 promotes KSR heterodimerization with RAF. As a result of KSR–RAF dimerization, KSR acts as an allosteric activator of RAF to enhance its catalytic activity in addition to its function as a scaffold connecting RAF with its substrate MEK [57]. On the other hand, according to the model proposed by Brennan and collaborators in 2011 [306], once KSR and BRAF dimerize, a conformational change in KSR exposes the MEK activation loop phosphorylation sites, S218 and S222. Once these sites in the MEK activation loop are exposed, an active RAF molecule from a second complex will phosphorylate MEK.

#### 4.2.2. IQGAP1/2 and 3

IQ motif-containing GTPase Activating Protein (IQGAP) 1 was first identified in 1994 by Weissbach and collaborators in a screen to discover novel matrix metalloproteases [368]. Nevertheless, it was not until 2004 that IQGAP was classified as a Ras/ERK pathway scaffold [324]. Mammalian IQGAP1, 2, and 3 are multidomain proteins of ~190 kDa [369,370,371]. IQGAPs exhibit different tissue expression patterns; while IQGAP1 is ubiquitous [368], IQGAP2 is mainly expressed in the liver and the gastro-intestinal and urogenital track [372,373], and IQGAP3 is found mainly in the brain, lung and testes [374]. IQGAP1 is cytoplasmic and mainly associated with the cytoskeleton, with particular enrichment in cell–cell contacts [375]. Distributions of IQGAP2 and IQGAP3 are not well described, but they have been noted throughout the cytoplasm and in cell–cell junctions, respectively [376,377]. A recent study reported IQGAP3 primarily in the nucleus [378].

IQGAP isoforms share a similar domain composition (Figure 5). Abundant protein-interacting motifs make them key players in numerous cellular processes. For example, more than 100 binding partners have been described for IQGAP1 [379]. At the N-termini of IQGAPs is a calponin homology domain (CHD) that binds cytoskeletal proteins, in particular F-actin [380]. This domain can also bind calmodulin/Ca^2+^ but at a lower affinity than to the IQ domain and in competition with F-actin [381]. The CHD is followed by a WW domain with two conserved Trp residues (W). WW domains specify proline-rich regions in binding partners. Interestingly, a major IQGAP WW domain-binding partner, ERK, apparently binds through a different mechanism [324]. Four tandem isoleucine/glutamine-containing (IQ) motifs are multipurpose domains that bind partners including calmodulin [381,382], the EGF receptor [383], MEK [384], BRAF [385] and the small GTPase RAP1 [386]. IQ domains are also important as they mediate the formation of IQGAP dimers and oligomers [387]. IQ domains participate in the interaction with phosphoinositide signaling elements such as phosphatidylinositol phosphate kinase type Iγ (PIPKIγ) at the leading edge. PIPKIγ binding favors the IQGAP open conformation to recruit the actin polymerization machinery and participate in cell motility [388]. At the C- terminus, there is the GAP-related domain (GRD), which defines IQGAP as a GTPase-activating protein (GAP). In spite of the family name and structural similarity to GAPs, IQGAPs do not promote GTP hydrolysis. A conserved arginine in GAPs is replaced by a threonine, which allows IQGAP1 to stabilize small GTPases in their GTP-bound states [369,389], similar to the Rho family GTPases RAC1 and CDC42 [389,390]. At its distal C-terminus, there is an RGCT (Ras GAP C-terminus) domain unique to IQGAP. This region binds diverse targets, such as phosphoinositides [388], E-cadherin [391] and β-catenin [392] among others. Overall, IQGAPs are unique complex proteins with exceptional binding capabilities, involving them in a broad variety of signaling pathways and cellular processes.

Despite the long list of interactors, a major role of IQGAP1 is as a scaffold for the RAS/ERK pathway [324]. IQGAP1 binds both MEK and ERK [384] and can also interact with BRAF and modulate its functions. Generally, IQGAP1–BRAF association is decreased by Ca^2+^/calmodulin binding to IQGAP1, as this interaction provokes a conformational change [393]. IQGAP1-bound BRAF has higher kinase activity than free BRAF; it is unclear whether binding to IQGAP1 promotes high BRAF activity or whether highly activated BRAF preferentially binds IQGAP [393]. EGF stimulates IQGAP1 binding to MEK1 and decreases binding to MEK2. It has been suggested that MEK1 binding to IQGAP1 promotes proliferation, whereas binding to MEK2 induces differentiation [394]. In contrast to MEK and BRAF, ERK1/2 constitutively bind IQGAP [324]. Overexpressed IQGAP3 also interacts with H-RAS [374], whereas IQGAP1 interacts with K-RAS [395,396]. IQGAP1 binding to EGFR modulates its activation [383], perhaps because the IQGAP1 scaffolded complex promotes EGFR phosphorylation by ERK when RAS is signaling from lipid raft domains [313]. IQGAP1 can also form complexes with other ERK pathway scaffolds such as β-arrestin-2 [397], MP1 [329], and KSR1 [332].

Under the control of RAC1/Cdc42 and antagonizing Ca^2+^/calmodulin signals, IQGAP1 participates in cell migration through focal adhesions and cytoskeleton dynamics by mediating co-localization of N-WASP with the Arp2/3 complex in lamellipodia which assemble to catalyze actin polymerization [398,399,400]. Stimuli that promote the formation of focal adhesions, such as PDGF, stimulate the assembly of IQGAP1 complexes with the focal adhesion proteins vinculin and paxillin [401]. During migration, a coordinated assembly and disassembly of focal adhesions is necessary. MP1 (MEK partner 1 aka LAMTOR3) plays a role in this process by anchoring MEK to late endosomes via p14 (LAMTOR2) [327]. Knock down of MP1 impairs migration due to the formation of abnormal focal adhesions that accumulate IQGAP1, suggesting that the interaction between IQGAP1, MP1 and the ERK cascade is crucial in this process [329]. IQGAP1 at the leading edge in migrating cells promotes migration in a RAC1- and Cdc42-dependent fashion. In response to integrin engagement, IQGAP1 forms a complex with filamin A to recruit RAC1GAP, which inactivates RAC1. Decreased expression of any of these three proteins results in uncontrolled formation of membrane protrusions and impairs directional cell migration [402]. F-actin crosslinking is key in the regulation of cytoskeletal processes such as the formation of focal adhesions and filopodia, essential for cellular migration [403]. IQGAP1 also aids in stabilizing F-actin crosslinking stimulated by high Cdc42 [404,405]. Interestingly, only IQGAP1 dimers, formed following Cdc42 stimulation, are competent for F-actin crosslinking and the generation of strong adherent junctions [406]. Recently, IQGAP1 has been found to participate as a scaffold in the Hippo pathway. IQGAP1 interacts with kinases MST2 and LATS1 to suppress their kinase activity. IQGAP1 is also a negative regulator of the non-canonical pro-apoptotic pathway [407].

IQGAP1, like KSR1, is not required for embryonic development. Mice lacking IQGAP1 are viable and fertile with gastric hyperplasia as a mild phenotype [408]. However, these mice are resistant to developing tumors induced by oncogenic H-RAS [325]. In the case of IQGAP2 knockout mice, they develop age-dependent hepatocellular carcinoma (HCC) with increased IQGAP1 expression [409]. These two phenotypes align with IQGAP1 pro-tumorigenic and IQGAP2 anti-tumorigenic roles. Indeed, in HCC, IQGAP1 protein and mRNA are increased, while IQGAP2 mRNA is downregulated [410,411] and IQGAP3 is high in gastric cancer [412]. Furthermore, the IQGAP1–MST2-LATS1 (Hippo pathway) complex is regulated by bile acid in HCC cells, likely to inhibit MST2-dependent apoptosis and cell transformation [407].

No definitive oncogenic mutations have been identified in IQGAPs in tumors; however, missense mutations have been identified that correlate with specific tumor types (Table 2), further suggesting that IQGAP1 participates in oncogenic processes [413]. Also, IQGAP gene amplifications have been found in tumors [414]. IQGAP1 was found overexpressed in lung, endometrial, ovarian, gastric, colon, and breast cancer, as well as hepatocellular carcinoma (HCC) and melanoma [415]. In fact, *IQGAP1* has been proposed as an oncogene because its overexpression can stimulate cell proliferation and favor transformation of human epithelial cells [415]. Similarly, downregulating IQGAP1 in MCF7 cells using siRNAs diminishes cell migration, proliferation and prevents tumor formation in xenograft mice [299,416].

#### 4.2.3. HPIP

HPIP (hematopoietic PBX (pre-B cell leukemia homeobox)-interacting protein or pre-B cell leukemia transcription factor-interacting protein (PBXIP1)) is a 731-amino acid protein that was discovered in 2000 in a yeast two-hybrid screen using PBX1a as bait. HPIP interacts with PBX transcription factors, negatively regulating PBX1-mediated hematopoiesis [417]. Although HPIP is predominantly cytoplasmic, specifically localizing on microtubules, this protein contains both a nuclear localization signal (NLS) and a nuclear export signal (NES) [418].

It was not until ten years after its discovery that HPIP was recognized as an ERK scaffold. HPIP associates with the estrogen receptor (ERα), Src and PI3K, which, in turn, activate ERK1/2 and AKT signaling pathways [419]. Thus, HPIP’s function as a scaffold protein is not exclusive to the RAS–ERK pathway. HPIP participates in signal transduction via transcription factors, C/EBPα and GATA1, ERα and ERβ, LEF1/ β-catenin and p53 as well as through tyrosine and serine/threonine protein kinases, FAK/Src, and TBK1 and CK1α, and it has also been implicated in the regulation by TGF-β. This great variety of interactors connect HPIP with processes including hematopoiesis, cell proliferation and tumorigenesis, germ cell proliferation, EMT, renal fibrosis, cell migration and invasion, and osteoarthritis. HPIP is widely expressed in different tissues, and it is considered a proto-oncogene since it is overexpressed in more than 15 types of cancer [420]. The only HPIP knockout mouse study revealed a defect in articular cartilage and bone development, but the lack of this protein protects against osteoarthritis [421].

In a recent study, HPIP was shown to facilitate the interaction of the canonical components of the RAS–ERK pathway in a phosphorylation-dependent manner. HPIP interacts with all the components of the pathway, from the EGFR to ERK, including all RAS isoforms (K-, N-, H-RAS), giving rise to two different functional complexes. First, HPIP associates with EGFR, SHC, Grb2, SOS1, RAS, RAF, and MEK to form complex 1. Subsequently, the scaffold leaves behind a few upstream components to bind ERK and make complex 2 with SHC, RAS, RAF, MEK, and ERK [422]. Further research is needed to fully understand the importance of this little-studied scaffold in health, disease, and potential therapeutic implications.

**Table 2 biomolecules-13-01555-t002:** Interactors, subcellular localization, tissue expression and mutations in RAS–ERK pathway scaffolds found in cancer. The information presented in this table has been retrieved from multiples sources, including string-db.org (accessed on 3 July 2023) [423], genecards.org (accessed on 3 July 2023) [424] and phosphosite.org (accessed on 3 July 2023) [425], as of July 2023. Data for subcellular localization obtained from GeneCards was collected from UniProtKB/Swiss-Prot, the Human Protein Atlas (HPA) and Gene Ontology (GO) (Confidence > 3). Tissue expression information is based on mRNA expression in normal human tissues from GTEx, Illumina, BioGPS, and SAGE (expression was considered ubiquitous when detected in over seven different tissues). The mutation frequency was obtained from TCGA (The Cancer Genome Atlas) in 15 types of cancer. The table lists up to the five most common cancer types with recurrent potential oncogenic missense mutations in each scaffold. The cancer types are listed in descending order of prevalence. Protein name is given in alphabetical order, and the gene name is in italics under the protein name.

Scaffold Name	RAS-ERK Pathway Interactors (Source: STRING and References)	Subcellular Localization (Source: GeneCards)	Tissue Expression (Source: GeneCards)	Mutations in Cancer (Source: Phospho-Site)	References
Archvillin/Supervillain *Spliced isoform of SVIL*	B-RAF, MEK, ERK	Cytosol, Cytoskeleton, PM, Nucleus	Ubiquitous	Stomach, bladder, endometrial, lung squamous, LUAD…	[426]
β-Arrestin-1 *ARRB1*	C-RAF, MEK, ERK	Nucleus, GA, Lysosome, Cytosol, PM	Ubiquitous	Lung squamous, Stomach, Endometrial, Bladder, CRC…	[427]
β-Arrestin-2 *ARRB2*	C-RAF, MEK, ERK	Cytosol, Nucleus, PM, Endosomes	Ubiquitous	Endometrial, LUAD, HNSCC, Stomach…	[427]
CaM *CALM1* *Calmodulin*	EGFR, RAS, RAF	Cytoskeleton, PM, Cytosol, Nucleus	Ubiquitous	LUAD, Endometrial, Bladder, Stomach, HNSCC.	[393]
CNK1 *CNKSR1* *Connector Enhancer Of Kinase Suppressor Of Ras 1*	RAS, RAF	Cytosol, PM, Nucleus	Highest in Muscle	Stomach, Endometrial, Bladder, CRC, LUAD…	[428]
CNK2/MAGUIN-1 *CNKSR2* *Connector Enhancer Of Kinase Suppressor Of Ras 1*	RAS, RAF	Cytosol, PM, Nucleus	Nervous system	Lung squamous, Endometrial, Stomach, CRC, LUAD…	[429]
DYRK1A *DYRK1A* *Dual Specificity Tyrosine Phosphorylation Regulated Kinase 1A*	RAS, B-RAF, MEK1	Nucleus, Cytosol, Cytoskeleton	Ubiquitous	Endometrial, Lung squamous, Stomach, CRC, HNSCC…	[430]
β-Dystroglycan-1 *DAG1* *Dystrophin-Associated Glycoprotein 1*	MEK, ERK	PM, Nucleus, Extracellular, Cytosol, GA, ER, Cytoskeleton	Ubiquitous	Stomach, CRC, Endometrial, LUAD, Lung squamous…	[431]
FHL1 *FHL1* *Four And A Half LIM Domains 1*	CRAF, MEK2, ERK2	Cytosol, Nucleus, PM	Ubiquitous	Endometrial, Stomach, LUAD, Lung squamous, Breast…	[432]
FLOT1 *FLOT1* *Flotillin 1*	RAF, MEK, ERK	Endosomes, PM, Lysosome, Extracellular, Cytosol, Cytoskeleton	Ubiquitous	Stomach, CRC, Lung squamous, Endometrial, HNSCC…	[433]
GIT1 *GIT1* *G Protein-Coupled Receptor Kinase Interactor 1*	MEK1, ERK1/2	Cytosol, Cytoskeleton, Mitochondria	Ubiquitous	Endometrial, bladder, Stomach, Lung squamous, kidney…	[434]
GAB1 *GAB1* *Grb2 Associated Binding Protein 1*	Grb2, SOS1	Cytosol	Ubiquitous	CRC, Endometrial, bladder, Lung squamous, Stomach…	[435]
Galectin1 *LGALS1* *Putative MAPK-Activating Protein PM12*	HRAS, CRAF	Cytosol, ER, Nucleus	Ubiquitous	HNSCC, Breast Lung squamous, CRC, LUAD…	[436]
Grb10 *GRB10* *Growth Factor Receptor-Bound Protein 10*	C-RAF, MEK	Cytosol, PM, Nucleus	Ubiquitous	Endometrial, LUAD, Stomach, Lung squamous, bladder…	[437]
HPIP/PBXIP1 *PBXIP1* *Hematopoietic PBX (Pre-B cell leukemia homeobox)-interacting protein*	EGFR, SHC, Grb2, SOS1, RAS, RAF, MEK, ERK	Cytosol, nucleus, Cytoskeleton	Ubiquitous	Stomach, Bladder, CRC, LUAD, Endometrial…	[419]
IQGAP1 *IQGAP1* *IQ Motif Containing GTPase Activating Protein 1*	EGFR, K-RAS, B-RAF, MEK, ERK	Cytosol, Nucleus, Cytoskeleton, PM, Focal Adhesions	Ubiquitous	Stomach, CRC, Endometrial, Lung squamous, HNSCC…	[324]
IQGAP2 *IQGAP2* *IQ Motif Containing GTPase Activating Protein 2*	RAS, B-RAF, MEK, ERK	Cytosol, Cytoskeleton, PM, Nucleus	Ubiquitous	Endometrial, CRC, LUAD, Lung squamous, Stomach	[413]
IQGAP3 *IQGAP3* *IQ Motif Containing GTPase Activating Protein 3*	H-RAS, B-RAF, MEK, ERK	Cytosol, Nucleus, PM, cell-cell junctions	Liver, lung, pancreas	Lung squamous, CRC, Endometrial, Stomach, Bladder…	[413]
KSR1 *KSR1* *Kinase Suppressor of Ras 1*	RAF, MEK, ERK	Cytosol, ER, PM	Ubiquitous	Endometrial, ovarian.	[333,334,335]
KSR2 *KSR2* *Kinase Suppressor of Ras 2*	RAF, MEK, ERK	Cytosol, PM	Nervous system	Stomach, Endometrial, LUAD, HNSCC, Lung squamous…	[336]
MERLIN *NF2* *Moesin-Ezrin-Radixin Like (MERLIN) Tumor Suppressor*	RAS, RAF	Cytosol, Cytoskeleton, PM, Nucleus, Mitochondria	Ubiquitous	Endometrial, Bladder, HNSCC, CRC, Lung squamous…	[438]
MORG1 *WDR83* *Mitogen-Activated Protein Kinase Organizer 1*	C-RAF, MEK, ERK	Endosomes, Nucleus	Kidney, Muscle, Nervous system	Endometrial, Bladder, Prostate, Lung squamous, CRC…	[328]
MP1 *LAMTOR3* *MEK Partner 1*	MEK1, ERK1, ERK2	Endosomes, Lysosome, PM, Extracellular	Ubiquitous	HNSCC, Glioblastoma.	[439,440]
Nucleophosmin *NPM1*	KRAS, RAF	Cytosol, Cytoskeleton, Nucleus,	Ubiquitous	Leukemia, LUAD, Endometrial, CRC, HNSCC…	[441]
Paxillin *PXN*	MEK, ERK	Cytosol, Focal Adhesions, Cytoskeleton, PM, Nucleus	Ubiquitous	Endometrial, Stomach, Lung squamous, HNSCC, Glioblastoma…	[300]
PEA15 *PEA15* *Phosphoprotein Enriched in Astrocytes, 15 KDa Proliferation And Apoptosis Adaptor Protein 15*	ERK, RSK	Cytosol, Cytoskeleton, Nucleus	Ubiquitous	Lung squamous, Endometrial, CRC, LUAD, HNSCC.	[148]
RGS12 *RGS12* *Regulator Of G Protein Signaling 12*	RAS, RAF, MEK2	Cytosol, Nucleus, PM	Nervous system, Lung, Kidney	Stomach, Lung squamous, bladder, Endometrial, CRC…	[442]
RGS14 *RGS14* *Regulator Of G Protein Signaling 14*	RAS, RAF	Nucleus, Cytoskeleton, Cytosol, PM	Ubiquitous	Endometrial, bladder, Stomach, CRC, LUAD…	[443]
SEF1 *IL17RD* *Interleukin-17 Receptor D*	MEK, ERK	GA, Nucleus, PM	Nervous system, heart, Kidney	Endometrial, CRC, Stomach, Leukemia, Breast…	[150]
SHOC2/SUR8 *SHOC2* *Leucine Rich Repeat Scaffold Protein*	RAS, RAF	Cytosol, Nucleus	Ubiquitous	Endometrial, Stomach, CRC, LUAD, HNSCC…	[444]

Abbreviations: LUAD = lung adenocarcinoma; HNSCC = head and neck squamous cell carcinoma; CRC = colorectal carcinoma; ER = endoplasmic reticulum; PM = plasma membrane; GA = Golgi apparatus. All data refers to Homo sapiens.

## 5. Pathway Inhibitors

Pharmacological small molecule inhibitors of great scientific utility have been developed for the enzymes in the cascade, while those targeting RAF and MEK dominate clinical strategies to block the pathway. Table 3 lists some of these compounds. Poor pharmacokinetics and other issues with small molecule kinase inhibitors have provoked the development of other strategies to inhibit high-impact targets. In this section we focus on several alternative strategies to manipulate the ERK pathway via docking site inhibitors, protein degraders for multiple pathway components, molecular glues, and optically activated MEK inhibitors. Some of these may also yield access to proteins considered undruggable such as transcription factors and scaffolds.

### 5.1. Classical Inhibitors

Many classical inhibitors for the different members of the cascade have been developed. Many of them are in clinical trials but only a few have received FDA approval, mostly those targeting RAF and MEK. Remarkably, two covalent inhibitors, sotorasib and adragasib, have been FDA-approved in the last couple of years to treat KRAS^G12C^ lung cancers (reviewed in [487]), and several promising clinical trials are currently in progress [30]. As of today, no ERK inhibitors are in clinical use, although several are in clinical trials [488]. Unfortunately, single-therapy approaches have led to the development of mechanisms of resistance. To overcome this issue, many combination strategies are being assessed and some have been already approved [489].

In contrast to many other kinases, ERK2 undergoes relatively small conformational changes in the active site upon activation. Solution measurements have found evidence for phosphorylated ERK2 (2P-ERK2) in two conformational states, R and L, which interconvert on a millisecond time scale [490]. Surprisingly, the L state resembles the active site of unphosphorylated ERK (0P-ERK2). A recent study found ERK inhibitors that interact with both states and ones that selectively interact with the R or L states. Inhibitors were noted that could shift the equilibrium towards the R or L states. As some ERK inhibitors have been shown to affect specific protein–protein interactions, this study provides new insights for inhibitor design [491].

Apart from the large number of classical small molecule inhibitors directed at the ATP site [492] (representative list in Table 4), some alternative inhibitors target protein–protein interactions. An example is the ERK dimerization inhibitor DEL-22379. This small molecule disrupts ERK dimerization by binding within the dimerization interface without affecting the ERK phosphorylation state [493,494]. Another strategy is to target ERK1/2 substrate docking sites [135,136], e.g., SF-3-030, which selectively interferes with a subset of ERK functions dependent on the ERK FXF docking site [495].

### 5.2. PROTAC Technology Applied to the ERK Pathway

As its name implies, Targeted Protein Degradation (TPD) covers several methods to induce the selective degradation of a specific molecule either via the proteasome, lysosome or autophagy system. One of these methods using the proteasome degradation system is PROTAC (Proteolysis-Targeting Chimera) [514]*,* initially developed in 2001 [515,516,517,518]. The strategy involves recruitment of an E3 ligase, usually VHL (von Hippel-Lindau VHL) or CRBN (Cereblon), in current versions of PROTACs, to the target protein to tag it for degradation. Two ligands, one that binds the target protein and a second that recruits an E3 ligase, are connected through a chemical linker. The target protein will be ubiquitinylated by the E3 and sent to the proteasome resulting in its near elimination [519]. Advantages of protein degradation versus classical small molecule inhibitors are many. PROTACs can reach the undruggable proteome as long as a suitable target ligand is available. Degradation prevents target protein accumulation, thus avoiding compensatory upregulation and drug resistance. PROTACs overcome the generation of resistant mutations under selective pressures or even the emergence of non-enzymatic functions of the targeted proteins, which might provoke a signaling rebound. With an appropriate high-specificity ligand, PROTAC may only target the pathogenic, mutated version of the protein, thus minimizing side effects. The catalytic nature allows them to be used at sub-stoichiometric concentrations to reduce toxicities, as the same PROTAC transiently binds the target protein, and after tagging it for degradation, it releases and binds another target protein. Protein degradation achieves an extended pharmacodynamic effect similar to that of covalent inhibitors, depending on the turnover rate of the target. Unlike small molecule inhibitors that function by an occupancy-driven mechanism, PROTACs possess an event-driven mode of action, which means that a brief interaction with the protein of interest is enough to trigger its degradation [519,520,521,522].

The major limitation of PROTACs continues to be poor oral bioavailability and low cell permeability due to their large sizes [523]. For this reason, it was not until 2019 that the first PROTAC progressed to clinical trials [524]. Although none are employed in the RAS–ERK pathway, a few FDA-approved therapies use degraders [525]. Nevertheless, due to their promising advantages, PROTACs are being developed to inhibit components within the RAS/ERK cascade, summarized here and in Table 5.

The first attempt to generate a PROTAC against KRASG12C was based on the inhibitor ARS1620, and a thalidomide analog, pomalidomide, an FDA-approved drug with high affinity for the ubiquitin ligase CRBN. The compound was named XY-4-88, but failed to induce endogenous KRASG12C degradation in cancer cells [526]. A parallel study generated LC-2, the first successful PROTAC against KRASG12C, which used MRTX849 as the parental inhibitor [527]. A new collection of KRASG12C-selective PROTACs were synthesized based on the AMG-510 covalent inhibitor and were effective, but the IC50s were above 10 µM [528].

The first series of compounds acting as PROTACs for BRAF were created based on the inhibitor rigosertib, which binds to RAF in the RAS binding domain (RBD) and should, therefore, bind RAF mutants as activating mutations are in the kinase domain [529], although this was not tested. P4B was the first effective PROTAC with selectivity for the BRAF mutant V600E. The ligand was based on the inhibitor BI882370 and provides degradative efficacy in the nanomolar range [530]. A year later a mutant-selective PROTAC based on vemurafenib, SJF-0628, was shown to work at nanomolar concentrations on all BRAF mutants tested, including those that showed resistance to vemurafenib, while it had no effect on the WT form [531]. Another BRAFV600E-selective PROTAC, CRBN(BRAF)-24, was based on PLX8394 [532]. In parallel, encorafenib-based PROTACs were developed but none of them resulted in degradation of the mutant BRAF, probably due to failure to recruit the ubiquitin ligase [533].

The first-in-class highly selective degrader of MEK1 and MEK2 was the PD0325901-based compound MS432. It was effective in colorectal cancer and melanoma cell proliferation, with good bioavailability in mice [534]. A few PROTAC compounds against MEK1 and MEK2 were developed based on allosteric MEK inhibitors, such as PD0325901 and refametinib, using VHL as the E3 ligase. They were assessed in cell proliferation and pERK levels to compare the compound with the parental inhibitor. Although these compounds had modest degradation efficiencies, two of them were able to completely suppress proliferation in melanoma cells [535]. Following the development of the first-in-class MEK degrader, the Lin laboratory also characterized three more, MS928, MS934 and MS910. Compared with previously reported VHL-recruiting degraders, these new compounds were more potent in preserving the high selectivity. Of these three degraders, MS934 appeared as the best MEK1/2 degrader for in vivo studies to date. In addition, combinations of MS934 with either BRAF or PI3K inhibitors provided a potent antiproliferative response in CRC and melanoma cells [536]. A new study has reported that MS934 collaterally degrades CRAF in KRAS mutant cells, which makes this compound a first-in-class dual PROTAC [537]. Remarkably, other PD0325901-based degraders did not cause CRAF collateral degradation. Therefore, MS934 inhibiting both CRAF and MEK would not be expected to induce resistance mechanisms via non-catalytic CRAF functions [537]. The aforementioned PROTACs were designed based on a diarylamine scaffold common to most of the inhibitors. Recently, MEK1/2 degraders were synthesized based on a non-diarylamine allosteric MEK1/2 inhibitor, which was a coumarin derivative [538]. From all of them, compound P6b is the first MEK PROTAC with a non-diarylamine scaffold showing a potent degradation effect in human cancer cells [539]. At this time, MEK appears to be the most promising target for PROTACs due to demonstrated functional efficacy in the nanomolar range, and the fact that the only known substrate of MEK is ERK minimizes toxicities by avoiding interference with other pathways.

**Table 5 biomolecules-13-01555-t005:** PROTAC-targeted protein degraders.

Name	Target (Protein of Interest)	Leading Compound (“Warhead”)	Linker	E3 Ubiquitin Ligase Target	E3 Recruiter (“Anchor”)	Reference
XY-4-88	KRAS^G12C^	ARS-1620	Cyclic tertiary amine-containing linker	CRBN	pomalidomide (thalidomide analog)	[526]
LC-2	KRAS^G12C^	MRTX849	Alkyl	VHL	VH032	[527]
III-2	KRAS^G12C^	AMG-510	polyethylene glycol (PEG)	VHL	(S,R,S)-AHPC (VH032-NH2) hydrochloride	[539]
P4B	BRAF^V600E^ BRAF^G466V^	BI882370	polyethylene glycol (PEG)	CRBN	pomalidomide (thalidomide analog)	[530]
SJF-0628	All mutant BRAF	vemurafenib	Rigid piperzine-based linker	VHL	VHL ligand (not specified)	[531]
CRBN(BRAF)-24	BRAF^V600E^	PLX8394	Cyclic tertiary amine-containing linker	CRBN	pomalidomide (thalidomide analog)	[532]
MS432	MEK1/2	PD0325901	alkyl	VHL	(S,R,S)-AHPC-Me	[534]
Several compounds	MEK1/2	Allosteric MEKi	Alkyl	VHL	VHL ligand (not specified)	[535]
MS928	MEK1/2	PD0325901	Alkyl	VHL	VH032	[536]
MS910	MEK1/2	PD0325901	polyethylene glycol (PEG)	CRBN	pomalidomide (thalidomide analog)	[536]
MS934	MEK1/2 CRAF	PD0325901	Alkyl	VHL	VH032	[536,537]
P6b	MEK 1/2	Substituted 3-benzylcoumarins allosteric MEKi	Alkyl	VHL	VHL ligand 22	[539]

CRBN, Cereblon. VHL, von Hippel–Lindau.

An enhanced PROTAC technology, known as CLIPTAC (In-Cell Click-Formed Proteolysis-Targeting Chimeras), involves the independent entry of the protein-binding ligand and the ubiquitin ligase anchor into the cell, where these precursors assemble and function as a regular PROTAC. This technique requires sequential treatment with each precursor because their assembly outside the cell will not result in protein degradation. Based on an ERK covalent inhibitor, this tool became the first degrader against ERK1/2 to show efficacy in cells [540]. On the other hand, ubiquibodies (uAbs) are newly engineered molecules able to mark proteins for degradation inside the cell. The uAbs consist of a universal E3 ligase fused to a synthetic binding protein such as a single-chain antibody fragment (scFv), a designed ankyrin repeat protein (DARPin), or a fibronectin type III (FN3) monobody [541]. Their main advantage is easy customizability by simply exchanging synthetic binding proteins with specificity for the target. The difference from regular PROTACs is that uAbs do not need to recruit ubiquitin ligase but they are the ligase themselves. This approach has been successfully applied for ERK degradation. In this case, the uAbs contain DARPins that bind to both phosphorylated and non-phosphorylated ERK1/2 [542].

### 5.3. Molecular Glues Targeting MEK

The natural products cyclosporin A and FK506 were among the earliest described as molecular glues by Schreiber in 1992 for their ability to promote interactions between two molecules that do not otherwise bind. Following this molecular glue concept, Simonetta (2019) [543] set out to develop a compound that would enhance a protein–protein interaction between a transcription factor and the E3 that normally degrades it. Subsequently, more molecular glues have been identified by high-throughput chemical screens and target validation [544,545]. This strategy involves a monovalent molecule able to stabilize a protein–protein interaction to modify the interactor’s surface to facilitate its binding with a new partner, even if they do not interact under normal conditions. Molecular glues can act as degraders when one of the binding proteins is an E3 ligase [546].

Because molecular glues are small molecules, unlike PROTACs, they should have superior membrane penetrability and oral bioavailability. On the other hand, PROTACs display a higher adaptability to target a protein of interest, while the original molecular glues had unknown ligands [523,546]. Recent studies are focusing on molecular glue optimization, as is the case for the novel MEKi trametiglue. Although approved for clinical use for a decade, trametinib was identified through phenotypic screens, and for this reason, the mechanism of action of trametinib remains incompletely understood. The Dar laboratory elucidated the binding mechanism of this small molecule inhibitor to its target by determining the crystal structure of trametinib-MEK-KSR. Trametinib simultaneously binds to MEK and its scaffold protein KSR, while it destabilizes the interaction of MEK with RAF. Trametiglue was created based on these notions, exhibiting an enhanced affinity for the MEK–KSR complex and demonstrating a better growth inhibition of KRAS mutant cells and lower signaling rebound than the original compound [547,548].

### 5.4. Optically Activated MEK Inhibitors

Coupling light with therapeutics offers spatial selectivity. The fundamental principle of this strategy is to administer a photoactive drug with low intrinsic (dark) toxicity to a patient followed by irradiation with light to activate it only in a defined region, thus minimizing on-target side effects and enhancing treatment precision. Three light-dependent pharmacological approaches have been developed as follows: photodynamic reactive therapy, irreversible photopharmacology (photodecaging), and reversible photopharmacology with a ligand that is active only in one structural state [549].

Light-dependent pharmacological approaches, specifically photo-decaging, have been proposed to prevent MEKi toxicities. The strategy consists of generating an inactive locked MEK inhibitor precursor, by linking a photo-cleavable protecting group (the ‘cage’) to the small molecule inhibitor. This drug protector is irreversibly cleaved from the MEK inhibitor by exposure to light, and then the inhibitor will bind MEK where it has been uncaged. This cutting-edge strategy has been carried out with the potent allosteric inhibitor PD0325901, which was previously discontinued in clinical trials due to its harmful effects which are presumably on-target. The optically activatable MEK1/2 inhibitors, named opti-MEKi, have shown promising efficacy in vivo in a melanoma xenograft zebrafish model [550]. Combinations of these approaches, e.g., photo-switchable linkers or light-activated degraders, are also in development [551].

## 6. Concluding Remarks

In conclusion, this review has offered an overview of pathway components and their functions, shedding light on the diverse range of scaffolds involved in pathway regulation. We have also emphasized emerging interventional strategies that hold promise for future advancements in this field. The intricate network and potential interventions discussed here underscore the exciting prospects for further research and therapeutic development within this pathway.

## Figures and Tables

**Figure 1 biomolecules-13-01555-f001:**
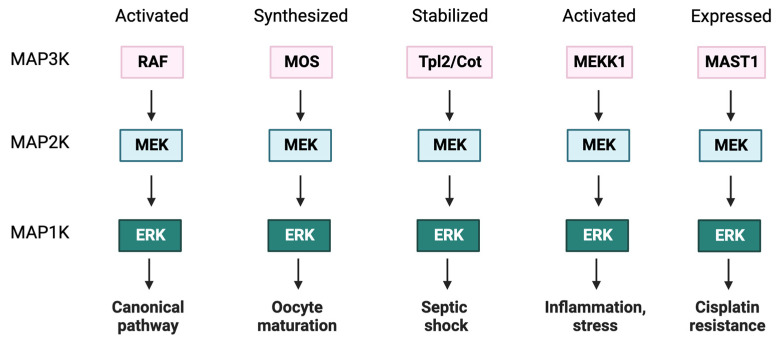
Activation of MEK by different upstream activators causes diverse outcomes. MEK can be activated by different MAP3Ks in a stimulus- and cell type-specific manner. The canonical pathway involves signal transduction through RAF. The synthesis of MOS induces oocyte maturation by direct phosphorylation of MEK and consequent activation of ERK. Upon stimulation, the kinase Tpl2/Cot is released from a ternary complex and is stabilized by phosphorylation of specific residues. The signaling module Tpl2/Cot-MEK-ERK is important in inflammation and immune response. MEKK1 is an upstream regulator of stress-responsive kinases, and it is also able to phosphorylate MEK1/2 to induce ERK activation in inflammatory settings. MAST1 initial or induced expression leads to cisplatin resistance.

**Figure 2 biomolecules-13-01555-f002:**
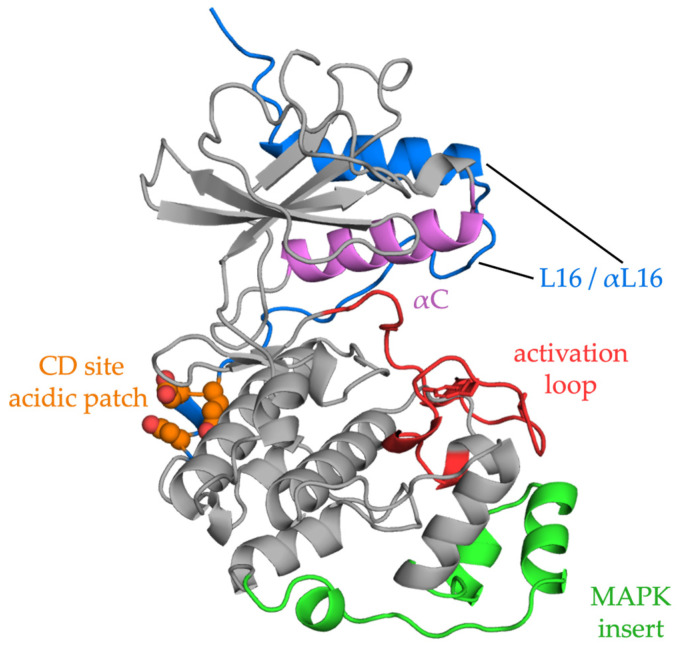
ERK2 structure. The ribbon diagram shows the two-domain structure of ERK2. Activation of the kinase occurs upon phosphorylation of the TEY motif in the activation loop (red), which lies in front of the active site. Phosphorylation causes a small domain closure, relative to other protein kinases. Reorganization of loop L16 (blue), part of a C-terminal extension that wraps up the back of the kinase, causes repositioning of alpha helix C (αC, magenta), allowing formation of a salt bridge between the glutamate in the helix to the essential lysine in beta strand 3 in the interior of the active site. The acidic residues that make up the CD domain are shown in orange. The FXF binding site lies between the MAPK insert (green) and the kinase core. Interactions through these two docking motifs may provoke conformational changes in ERK2 [115,116,117,118].

**Figure 3 biomolecules-13-01555-f003:**
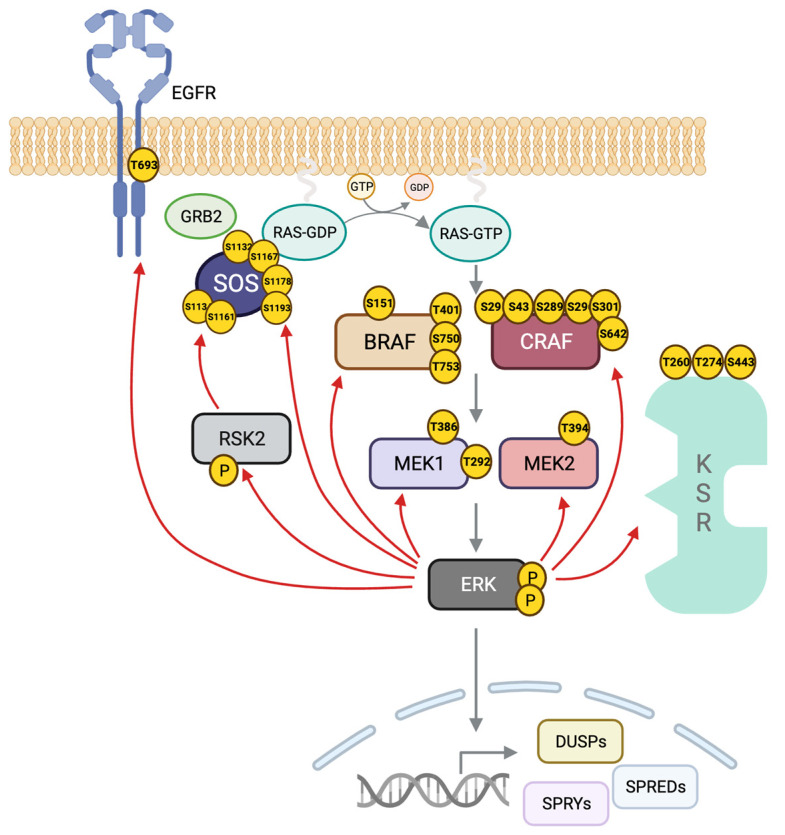
ERK negative feedback loops. ERK phosphorylates MEK1 at T292, thus impeding dimerization with MEK2 and PAK1-mediated phosphorylation of MEK1 in S298. ERK also phosphorylates T386 in MEK1 and the homolog residue in MEK2, T394. In CRAF, there are five ERK phospho-sites throughout the protein as follows: S29 and S43 in the N-terminus, S642 at the C-terminus, and S289, S296 and S301 at the flexible hinge between the regulatory and catalytic domains. The result is impaired localization at the plasma membrane, loss of CRAF–RAS interaction, and enhanced access for PP2A facilitating CRAF dephosphorylation. In BRAF, ERK phosphorylates S151 next to the RAS binding domain (RBD) of RAF, impairing RAS–BRAF interaction. ERK phosphorylates T401, S750 and T753 in BRAF, interfering with BRAF–CRAF dimerization. Pathway activation induces phosphorylation of KSR1 on T260, T274, S320 and S443 by ERK impairing dimerization with RAF and localization in the plasma membrane. ERK directly phosphorylates SOS on S1132, S1167, S1178 and S1193 and indirectly in S1134 and S1161 through RSK2. ERK phosphorylates the EGFR on T693 in the mature form of the receptor.

**Figure 4 biomolecules-13-01555-f004:**
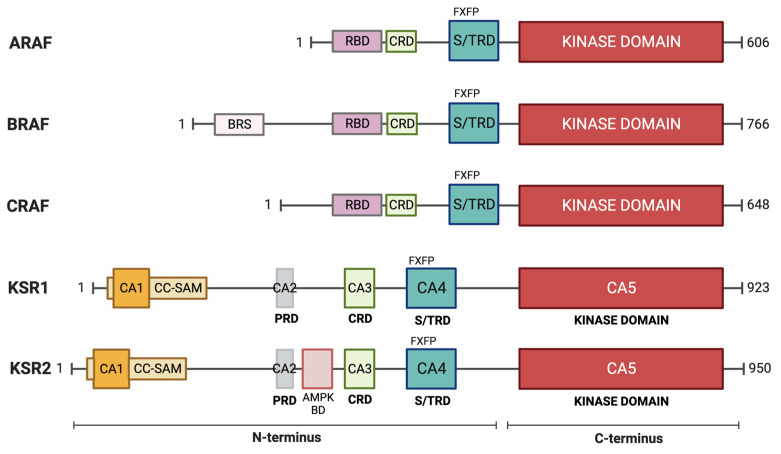
Schematic of the RAF and KSR protein structures. Domain organization of RAF consists of a RAS binding domain (RBD) followed by a cysteine-rich domain (CRD), both in the N-terminal regulatory region of the protein, and the C-terminal part is a kinase domain. BRAF presents an additional sequence at its N-terminal region, the BRAF specific sequence (BRS). KSR1 and KSR2 are composed of the same five conserved areas, CA1-CA5 [333]. CA1 to CA4 comprise the N-terminal region and constitute the regulatory domain, while CA5 is the pseudokinase region at the C-terminus. CA1 has a coiled-coil fused to a sterile α-motif (CC-SAM, not present in RAF) implicated in KSR1 localization at membrane ruffles [339]; this is not as clear for KSR2 since it has been considerably less studied. CA1, together with CA5, participates in B–RAF binding in a CC-SAM-independent manner [261,340]. CA2 is a proline-rich region, and it is believed to function as an SH3 binding domain. CA3, a cysteine-rich domain (CRD), is a lipid binding domain also involved in KSR subcellular localization, and it is essential to target KSR to the plasma membrane upon RAS activation [341,342,343,344]. CA4 is a serine/threonine-rich region which includes the FXFP docking motif for ERK binding [126,133,345]. Lastly, the CA5 kinase domain contains the MEK [346] and RAF [347] binding domains. Colors indicate structural similarities between family members including the cysteine-rich domain (CRD), serine/threonine-rich domain (S/TRD) and kinase domain.

**Figure 5 biomolecules-13-01555-f005:**
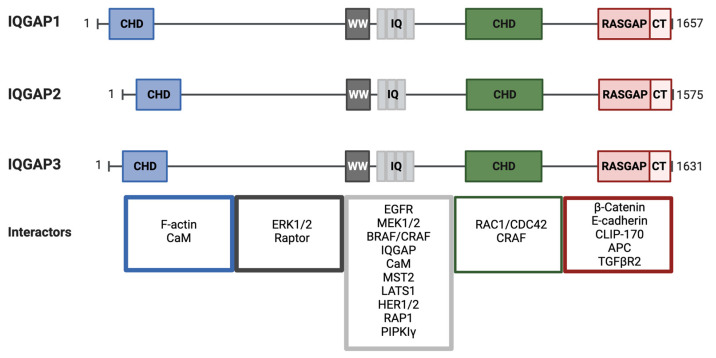
Schematic of IQGAP1/2 and 3 domain organization and interactors. IQGAPs are divided into five conserved domains, from N- to C-terminus. CHD, calponin homology domain; WW domain, a poly-proline protein–protein domain with two conserved Trp (W) residues; IQ domain, with four repeats (three for IQGAP2) of isoleucine- and glutamine-containing motifs; GRD, GAP-related domain; and RGCT, RAS GAP C-terminus domain. Some representative binding partners for each domain are organized and displayed in the color-coded boxes.

**Table 1 biomolecules-13-01555-t001:** Dissociation constant between different pairs of pathways components. K_d_ values have been calculated with data obtained from different in vitro methods as indicated in the corresponding reference.

Interaction	K_d_	Reference
HRAS (GDP and GTP-bound) dimer		[280]
in vitro (in solution)	NB	
in vitro (membrane-associated)	500 µM	
KRAS (GTP-bound) dimer		
in vitro (in solution)	0.77 µM	[281]
in vitro (membrane-associated)	530 µM	[282]
BRAF L487G dimer in vitro (in solution)	2 µM	[57]
MEK1 homodimers		
in vitro (in solution)	1.5 µM	[283]
Computational prediction	1.2 × 10^−11^ M^−1^	[83]
MEK2 homodimers Computational prediction	3.7 × 10^−9^ M^−1^	[83]
MEK1-MEK2 dimers Computational prediction	1.1 × 10^−11^ M^−1^	[83]
HRAS-GTP and ARAF-RBD	76.6 nM	[284]
HRAS-GTP and BRAF-RBD	11.2 nM	
HRAS-GTP and CRAF-RBD in vitro (in solution)	21.2 nM	
RAS-GTP and CRAF-RBD in vitro (in solution) (physiological ionic strength)	130 nM	[285]
KRAS-GTP and CRAF-RBDGRD in vitro (in solution)	152 nM	[50]
MEK1 and BRAF in vitro (in solution)	43 nM	[286]
	55 nM	[287]
	51 nM	[288]
MEK1 and CRAF in vitro (in solution)	38 nM	[287]
ppMEK1-ERK1	58 nM	[289]
ppMEK1-ERK2	340 nM (K_m_)	[290]
ERK2 and DUSP6 in vitro (in solution)	1.1 µM	[163]
ERK2 and RSK1	150 nM	[130]
ppERK2 and RSK1 in vitro (in solution)	120 nM	
ERK2 and ELK1	250 nM	[130]
ppERK2 and ELK1 in vitro (in solution)	>10 µM	
ERK2 and cFos	970 nM	[130]
ppERK2 and cFos in vitro (in solution)	1.3 µM	
ERK dimer in vitro (in solution)		[291]
Phosphorylated	7.5 nM	
Unphosphorylated	20 µM	

**Table 3 biomolecules-13-01555-t003:** List of representative allosteric or ATP-competitive MEK inhibitors. Some of the earliest inhibitors present major off-target effects, such as PD98059 and U0126 [445]. Among the most selective is PD0325901. The two values MEK1 and MEK2, respectively. (ND = not determined).

Name	IC_50_	Type	Target	Reported	Reference
PD98059	2–7 µM 50 µM	non-ATP-competitive	MEK1/2	1995	[446,447]
U0126	72 nM 58 nM	non-ATP-competitive	MEK1/2	1998	[448,449]
CI-1040/ PD184352	17 nM	non-ATP-competitive	MEK1	1999	[450,451]
Mirdametinib/PD0325901	0.33 nM	non-ATP-competitive	MEK1/2	2004	[451,452]
PD184161	10–100 nM	non-ATP-competitive	MEK1/2	2006	[453]
Binimetinib/ MEK162 (MEKTOVI)	12 nM	non-ATP-competitive	MEK1/2	2006 (FDA-approved: 2018)	[454]
Selumetinib/AZD6244/ARRY-142886 (KOSELUGO)	14 nM	non-ATP-competitive	MEK1	2007 (FDA-approved: 2020)	[455]
Refametinib/RDEA-119/BAY 869766	19 nM 47 nM	non-ATP-competitive	MEK1/2	2009	[456]
RO4987655/CH4987655/ RG7167	5.2 nM	non-ATP-competitive	MEK1	2009	[457,458]
AZD8330/ARRY-704	7 nM	non-ATP-competitive	MEK1/2	2009	[459]
E6201	50 nM	ATP-competitive	MEK1	2009	[460]
Pimasertib/ AS703026/MSC1936369B	ND	non-ATP-competitive	MEK1/2	2010	[461]
WX-554/UCB1366554	4.7 nM 10.7 nM	non-ATP-competitive	MEK1/2	2010	[462]
RO5068760	25 nM	non-ATP-competitive	MEK1	2010	[463]
G-573	16 nM	non-ATP-competitive	MEK	2010	[464]
TAK-733	3.2 nM	non-ATP-competitive	MEK1/2	2011	[465]
CIP-137401	21 nM	non-ATP-competitive	MEK1	2011	[466]
Trametinib/GSK1120212/JTP-74057 (MEKINIST)	2 nM	non-ATP-competitive	MEK1/2	2011 (FDA-approved: 2013)	[467,468]
Avutometinib/RO5126766/ RG-7304/ CH-5126766, CKI-27, R-7304/VS-6766	8.2 nM 56 nM 160 nM 190 nM	non-ATP-competitive	BRAF^V600E^ CRAF MEK BRAF	2012	[469]
Cobimetinib/ GDC-0973 (COTELLIC)	4.2 nM	non-ATP-competitive	MEK1	2012 (FDA-approved: 2015)	[470]
G-894	13 nM	non-ATP-competitive	MEK1/2	2012	[471]
GDC-0623/RG 7421	0.13 nM	non-ATP-competitive	MEK1	2013	[472]
Tunlametinib/HL-085	1.9–10 nM	non-ATP-competitive	MEK1	2013	[473,474]
CZ0775	>10 μM	non-ATP-competitive	MEK1/2	2013	[475]
BI-847325	25/4 nM 25/15 nM	ATP-competitive	MEK1/2 Aurora kinase A/C	2015	[476]
EBI-1051	3.9 nM	non-ATP-competitive	MEK	2018	[477]
SC-1-151	26 nM	non-ATP-competitive	MEK1/2/5	2018	[478]
DPS-2	<5 µM	non-ATP-competitive	MEK/ERK PI3K/AKT	2019	[479]
KZ-001	7.4 nM	non-ATP-competitive	MEK1	2019	[480]
URML-3881	30 nM	non-ATP-competitive	MEK1/2	2019	[481]
FCN-159	8.8 nM	non-ATP-competitive	MEK1/2	2020	[482,483]
MAP855	3nM EC_50_ 5 nM (pERK)	ATP-competitive	MEK1/2	2022	[484]
ABM-168	<30 nM	non-ATP-competitive	MEK1/2	2023	[485]
DS03090629	74 nM	ATP-competitive	MEK1/2	2023	[486]

**Table 4 biomolecules-13-01555-t004:** List of representative ATP-competitive ERK inhibitors. ND = not determined; NA = not applicable.

Name	IC_50_	Type	Target	Reported	Reference
FR 180204	0.31 µM 0.14 µM	ATP-competitive	ERK1/2	2005	[496]
VTX-11e (Vertex-11e)	K_I_ < 2 nM	ATP-competitive	ERK1/2	2009	[497]
AEZS-136	50 nM	ATP-competitive	ERK1/2 PI3K	2012	[498]
SCH772984	4 nM 1 nM	ATP-competitive	ERK1/2	2013	[499]
Ravoxertinib (GDC-0994)	1.1 nM 0.3 nM	ATP-competitive	ERK1/2	2014	[500]
Ulixertinib (BVD-523, VRT752271)	<0.3 nM (ERK2)	ATP-competitive	ERK1/2	2015	[501,502]
LTT462	ND	ATP-competitive	ERK1/2	2016	NA (ClinicalTrials.gov Identifier: NCT02711345)
Temuterkib (LY3214996)	5 nM	ATP-competitive	ERK1/2	2017	[503,504]
KO-947	10 nM	ATP-competitive	ERK1/2	2017	[505]
ASN007 (ERK-IN-3)	Low nM	ATP-competitive	ERK1/2	2018	[506]
MK-8353 (SCH900353)	20 nM 7 nM	ATP-competitive	ERK1/2	2018	[507,508]
CC-90003	10–20 nM	ATP-competitive	ERK1/2	2019	[509]
AZD0364	6 nM	ATP-competitive	ERK1/2	2019	[510]
HH2710	4.5 nM 4.9 nM	ATP-competitive	ERK1/2	2020	[511]
JSI-1187	K_I_ < 1 nM	ATP-competitive	ERK1/2	2020	[512]
ASTX029	3 nM 2.7 nM	ATP-competitive	ERK1/2	2021	[513]
HMPL-295	ND	ND	ERK1/2	2021	NA (ClinicalTrials.gov Identifier: NCT04908046)

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
