# Peer review of "Navigating the ERK1/2 MAPK Cascade"

_biomolecules, 2023, doi:10.3390/biom13101555_

Round 1

Reviewer 1 Report

The Ras-ERK signaling is fundamental to cell biology. Aberrant activation of this pathway induces many human diseases, particularly cancers. In this review, Cobb and colleague summarised the latest proceedings of research on this signaling pathway with a focus on ERK1/2, and highlighted novel approaches to target this signaling pathway for disease treatment, which should have important implications for future understanding underlying regulatory mechanism as well as developing effective therapeutics. Overall, this manuscript is comprehensive and very confluent, and suitable for Biolmolecules. I have only one minor suggestion, if authors can include the recent development of ERK inhibitors in this manuscript, it would improve its integrity. 

Author Response

Thank you for your time reviewing our manuscript, we really appreciate your suggestion. We have now included a table with some representative ERK inhibitors in section 5.1 as table 4.

Reviewer 2 Report

This extensive review of ERK and MAPK is an excellent up-to-date survey of the literature. It should serve as a valuable resource to both the MAPK signaling expert and those peripherally interested. The section on MAPK scaffolds is particularly notable. The tables of the effector interaction and their affinities and the table of the various scaffold and their characteristics as well the table on MEK inhibitors should be broadly useful. The context dependent signaling of ERK, and its subcellular regulation was an important point of emphasis. 

The overall impression is that while these are an extensive collection of observations from many labs study the many and varied modifiers of ERK signaling, we still lack a comprehensive understanding of their role in the context of particular biological effects (e.g., in glucose-regulated insulin secretion of beta cells).

The section on inhibitors, PROTACs and other methods of targeting ERK and other MAPK pathway components is welcome. Even if the work on some of these areas is in its infancy, it is important to make interested investigators aware of these efforts.

Two minor issues were noted:

On line 683-684 the authors write, “…reported that ATP-binding is essential for RAF activation, although how KSR1 binds ATP was not addressed [303,356]. Here, I believe the authors meant to write “KSR1” in place of “Raf”. For example, the paper from the Barbacid lab (ref 356) suggests the ATP binding to KSR1 is important for its function. 

On line 876-877 the authors write, “…while those targeting RAF and MEK dominant clinical strategies to block the pathway.” Do they mean “dominate”?

Author Response

We appreciate your time reading and providing feedback on our manuscript. Certainly, both observations were right. It is now corrected in the revised version of the manuscript.

Reviewer 3 Report

This manuscript is an excellent, fairly comprehensive review of the ERK MAP Kinases. Other reviews are referenced that cover areas of ERK signaling that are not reviewed in depth in this manuscript. Upstream signaling components are discussed, as is the role of these activators (Ras and Raf family members) in cancer, with detailed information on functional differences between family members and specific mutations and their effects. The role of scaffold proteins is discussed, with specific examples and a comprehensive table. The mechanisms of activation and regulation of the individual members of the ERK pathway are discussed in depth. The role of ERK in signal transduction through phosphorylation of substrates is well-covered. Finally, there is an up-to-date discussion on pathway inhibitors and where they stand in clinical trials and FDA approval. The figures and tables contribute to the understanding of the text and the comprehensive nature of the review.

A couple of minor points:

Line 166: there are two references that seem out of place or are not numbered (Seger, 1992; Crews, 1992). Please fix.

Lines 249 and 250: It is unclear why these lines are in red font.

Author Response

Thank you for your time reviewing our manuscript and your observations, we have fixed both issues in the revised version.

Reviewer 4 Report

The ERK1/2 kinases have critical roles in many signal transduction pathways, they have been extensively studied and a vast information about them is currently available. This review is therefore timely in updating the field on what is now known about these protein kinases, particularly as therapeutic target. The paper is clearly written and in principle publication is recommended for publication provided that the authors can address the minor comments outlined below.

Section ERK1/2 in cancer should be included in section 2.3 ERK, without heading.

The names of the proteins should be consistent throughout the manuscript, one example is TPL2 or Tpl2

There is one sentence (Figure 2 legend) in red, please correct

On line 275 there is a mistake in the format of the references

Author Response

Thank you for your time reviewing our manuscript and providing valuable feedback. We apologized for the confusion regarding the subheadings. There was a mistake where MEK and the following section ERK were both numbered as 2.3. Now, ERK is section 2.4 and we have accordingly updated the subheadings. Given the extensive review of ERK compared to the upstream components of the pathway, we found it necessary to divide this section into subsections to improve the comprehension and flow. The rest of the concerns have been fixed in the revised version of the manuscript.
